# ImageRAG: Dynamic Image Retrieval for Reference-Guided Image Generation

**Rotem Shalev-Arkushin** [1]   **Rinon Gal** [1,2]   **Amit H. Bermano** [1]   **Ohad Fried** [3]

[1] Tel-Aviv University   [2] NVIDIA   [3] Reichman University

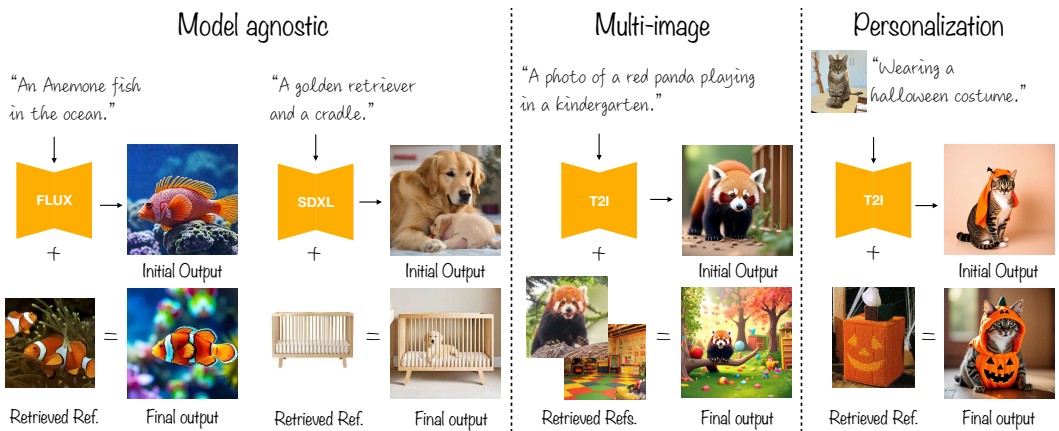

Figure 1: Using image references broadens the generation capabilities of image generation models. Given a text prompt, our method, *ImageRAG*, dynamically retrieves relevant images and provides them to a text-to-image model as references (*Retrieved Ref.*). *ImageRAG* works with different models, e.g. FLUX and SDXL (left), and OmniGen (right), and with different controls, e.g. text (left, middle), and personalization (right).

## Abstract

While recent generative models synthesize high-quality visual content, they still struggle with generating rare or fine-grained concepts. To address this challenge, we explore the usage of Retrieval-Augmented Generation (RAG) for image generation, and introduce *ImageRAG*, a training-free method for rare concept generation. Using a Vision Language Model (VLM), *ImageRAG* identifies generation gaps between an input prompt and a generated image dynamically, retrieves relevant images, and uses them as context to guide the generation process. Prior approaches that use retrieved images require training models specifically for retrieval-based generation. In contrast, *ImageRAG* leverages existing image conditioning models, and does not require RAG-specific training. We demonstrate our approach is highly adaptable through evaluation over different backbones, including models trained to receive image inputs and models augmented with a post-training image-prompt adapter. Through extensive quantitative, qualitative, and subjective evaluation, we show that incorporating retrieved references consistently improves the generation abilities of rare and fine-grained concepts across three datasets and three generative models. Our project page is available at: `https://rotem-shalev.github.io/ImageRAG`

## 1 Introduction

Deep generative models (Ho et al., 2020; Rombach et al., 2022; Dhariwal & Nichol, 2021; Labs, 2023) have revolutionized image generation, offering high-quality, diverse, and realistic visual content synthesis. They enable text-to-image generation as well as a wide range of tasks, from layout-based

synthesis to image editing and style transfer (Avrahami et al., 2022; Hertz et al., 2022; Mokady et al., 2023; Avrahami et al., 2023b; Zhang et al., 2023; Brooks et al., 2023; Nitzan et al., 2024). These large models require great amounts of training data, substantial training durations, and extensive computational resources. As a result, contemporary text-to-image (T2I) models, that are limited to the data they were trained on, struggle with generating user-specific concepts or updated content. Specifically, they have difficulty with generating rare concepts, stylized content, or fine-grained categories (e.g., a specific bird species, as in Fig. 2, left), even if they were trained on images containing them (Samuel et al., 2024b; Haviv et al., 2024). In these cases, diffusion models tend to "hallucinate", and potentially generate content unrelated to the textual prompt (see Fig. 2, right). To

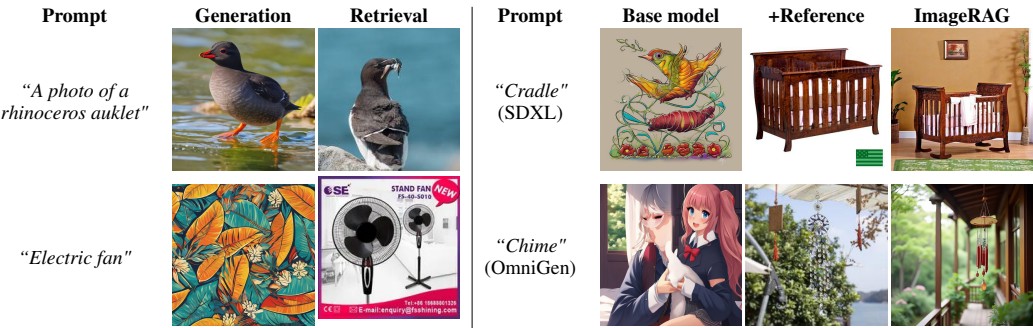

Figure 2: **Left: generation vs. retrieval.** Although some generative models, such as SDXL, use CLIP as a text encoder, they sometimes fail to generate concepts ('Generation' column) that are retrieved successfully using CLIP ('Retrieval' column). **Right: Hallucinations.** When models do not know the meaning of a prompt, they may "hallucinate" and generate unrelated images ('Base model' column). By applying our method to retrieve and utilize relevant references ('+Reference' column), the base models can generate appropriate images ('ImageRAG' column).

tackle these problems, several approaches have been proposed for personalized (Gal et al., 2023; Ruiz et al., 2023; Voynov et al., 2023; Arar et al., 2024), stylized (Hu et al., 2021; Li et al., 2024a; Zhang et al., 2024), or rare concept (Samuel et al., 2024b; Pan et al., 2025) generation. Most approaches, however, require training or specialized optimization techniques for each new concept.

We observe that similar problems exist with text generation using Large Language Models (LLMs). LLMs struggle with generating text based on real-world facts, proprietary or updated data, and tend to hallucinate when lacking sufficient knowledge (Brown, 2020; Ji et al., 2023). To solve these problems with LLMs, Retrieval Augmented Generation (RAG) (Lewis et al., 2020) has been proposed. RAG dynamically retrieves the most relevant information from external data sources given a query, and supplies it to an LLM as context input, enabling contextually accurate and task-specific responses. Investigating this idea for images, we note that previous works employing image retrieval for better image generation (Chen et al., 2022; Sheynin et al., 2022; Blattmann et al., 2022; Hu et al., 2024), train models specifically for the task, hindering wide applicability. A recent work (Lyu et al., 2025) requires training of a specific retrieval module per base generative model. In contrast, we propose *ImageRAG*, a method that dynamically retrieves and provides images as references to pretrained models, to enhance their generation capabilities, and does not require any additional training. Instead, we use existing generative models in the same vein as the common use of LLMs, and retrieve reference images during sampling for guided generation, by leveraging existing Vision-Language Models (VLMs). We note that in the visual domain, the context is more limited compared to the language case. Hence, to apply RAG for image generation, we cannot simply provide references for all the concepts in a prompt. Therefore, we need to decide which images to use as context, how to retrieve them, and how to use the retrieved images, to successfully generate a required prompt.

In this work, we address these questions by offering a novel method that dynamically identifies relevant and useful examples given a prompt, and uses them as references to guide the model toward generating the required result. Leveraging the abilities of T2I models to produce many concepts, we only pass concepts that the models struggle to generate, focusing on generation gaps. To understand what the challenging concepts are, we propose a novel method that applies a guided multimodal chain-of-thought (CoT) process using a VLM. By encouraging step-by-step reasoning, CoT (Wei et al., 2022a; Yang et al., 2023) helps VLMs reach more reliable and consistent conclusions by

decomposing complex tasks into explicit intermediate steps that reduce errors. Specifically, we first generate an initial image and then iteratively prompt a VLM to assess the alignment between the image and prompt, identify missing visual components, and suggest complementary concepts. We then use these concepts to retrieve reference images that guide subsequent generation.

Our approach is not related to a specific T2I model, and can be applied to different base models. To demonstrate it, we apply *ImageRAG* to three models: Omnigen (Xiao et al., 2024), SDXL (Podell et al., 2024)+IP-adapter (Ye et al., 2023), and FLUX (Labs, 2023)+OminiControl (Tan et al., 2024). We perform quantitative, qualitative, and human evaluations of our method with these models, and show that ImageRAG enhances their rare and fine-grained concept generation capabilities. These results indicate that the image generation community could benefit from adopting RAG for class or task-specific generation during sampling time.

## 2 RELATED WORK

**In-context learning (ICL)** has emerged as a powerful paradigm in which large language models (LLMs) are capable of performing new tasks without additional fine-tuning (Brown, 2020). By providing a few examples or relevant context directly in the input prompt, ICL enables models to infer the desired task and generate appropriate outputs. Despite its flexibility, ICL is limited by the finite context window of the model, making the selection of relevant and concise context critical for optimal performance. Recently, visual ICL presented promising results (Gu et al., 2024; Wang et al., 2023; Xiao et al., 2024; Najdenkoska et al., 2024; Sun et al., 2024). Visual ICL has mostly been explored in the context of learning from analogies (Gu et al., 2024; Wang et al., 2023; Xiao et al., 2024; Nguyen et al., 2024). However, the ability of learning from single examples has also been researched with multimodal generative models that allow images as input (Xiao et al., 2024; Sun et al., 2024; Wang et al., 2024b). Such models allow image prompting and facilitate exploring RAG for image generation.

**Retrieval Augmented Generation (RAG)** Lewis et al. (2020) is a method for improving the generation abilities of a pretrained model without additional training, by dynamically retrieving and supplying information as context through text prompts. For each given query, relevant information is retrieved from an external database, and supplied to the model for improved generation that relies on it as context. While RAG has been greatly explored for text generation tasks and applied over multiple pretrained LLMs (Lewis et al., 2020; Gao et al., 2023; Ram et al., 2023; Borgeaud et al., 2022; Li et al., 2024b; Zhang et al., 2025), it has yet to be explored for enhancing pretrained image generation models' capabilities. Some previous work used nearest-neighbor image retrieval to improve image generation (Sheynin et al., 2022; Blattmann et al., 2022; Chen et al., 2022; Lyu et al., 2025) or for editing-guidance (Hu et al., 2024; Sanguigni et al., 2025), however they either train models specifically for retrieval-aided generation or require a retrieval-module training per-model. Unlike them, our method leverages pretrained models and does not require additional training. A recent work, Yuan et al. (2025), utilizes an LLM for prompt decomposition to concepts and retrieves references representing each of them. Then, they generate a layout with all the concepts to assert all of them are generated. Unlike our method, they retrieve all the concepts in a prompt, ignoring prior knowledge of the model. Moreover, while using a layout ensures all concepts are present in the result, this strategy may constrain the generation abilities to isolated concepts without interaction between them. For example, their method may struggle to generate a concept in a style or interacting concepts.

**Text-to-image generation** advanced greatly with the introduction of diffusion models (Ho et al., 2020), which produce high-quality and diverse images of a wide range of concepts (Dhariwal & Nichol, 2021; Rombach et al., 2022; Podell et al., 2024; Xiao et al., 2024). However, they struggle with rare concepts and cannot generate user-specific concepts without additional training or optimization.

**Personalization** works generate images of a user-specific concept. However, they often require an optimization process for learning each new concept (Nitzan et al., 2022; Gal et al., 2023; Ruiz et al., 2023; Arar et al., 2024; Alaluf et al., 2023; Voynov et al., 2023; Avrahami et al., 2023a; Kumari et al., 2023). To mitigate this challenge, recent works train image-encoders that allow prompting existing pretrained generative models with images (Ye et al., 2023; Wang et al., 2024a).

**Rare Concept Generation** focuses on image generation of uncommon concepts. Samuel et al. (2024a;b) explored generating rare concepts using a few examples of each rare concept to optimize

seeds that produce images similar to the references. However, in addition to the requirement of an optimization process per new concept, these works do not address the questions of how to find and choose the reference images. Pan et al. (2025) suggests a parameter-efficient fine-tuning approach over full datasets.

# 3 METHOD

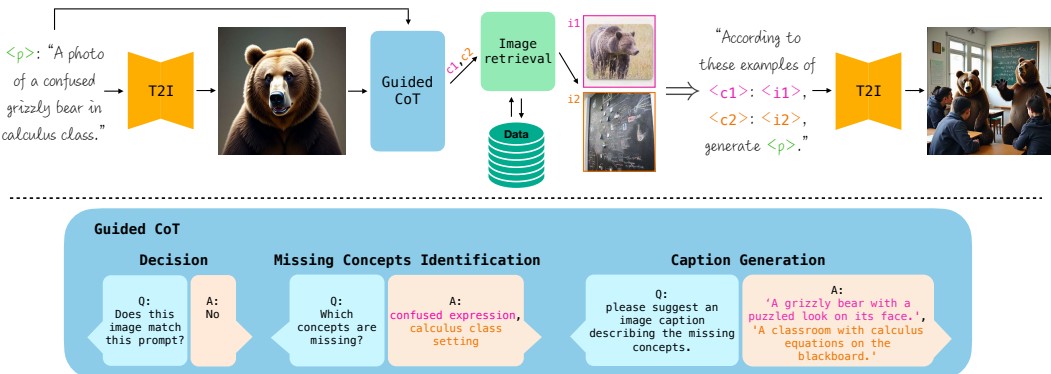

Figure 3: **Top**: a high-level overview of our method. Given a text prompt $<p>$, we generate an initial image using a text-to-image (T2I) model. Then, using a guided CoT process, we generate retrieval-captions $<c_j>$, retrieve images from an external database for each caption $<i_j>$, and use them as references to the model for better generation. **Bottom**: the guided Chain-of-Thought (CoT) process. We use a VLM to decide if the initial image matches the given prompt. If not, we ask it to list the missing concepts, and to create a caption that could be used to retrieve appropriate examples for each of the missing concepts.

Our goal is to increase the robustness of T2I models, particularly with rare concepts and fine-grained categories, which they struggle to generate. To do so, we investigate a retrieval-augmented generation (RAG) approach, through which we dynamically select images that can provide the model with missing visual cues. Importantly, we focus on models that were not trained for RAG, and show that existing image conditioning tools can be leveraged to support RAG post-hoc. As depicted in Fig. 3, given a text prompt and a T2I generative model, we start by generating an image with the given prompt. Then, we employ a guided chain-of-thought (CoT) reasoning process, through which a VLM decides whether the image aligns with the prompt, and identifies gaps in it if not. If gaps were identified, we aim to retrieve images representing missing concepts that would help fill these gaps. These images are then used as context to guide the T2I model toward better alignment with the prompt. In the following section, we provide a detailed description of our method.

## 3.1 GUIDED CHAIN-OF-THOUGHT (COT)

To identify missing concepts in an image and retrieve relevant references, we employ a guided CoT process with a VLM (depicted in Fig. 3, bottom). Since currently the amount of images we can pass to an image generation model is limited, we cannot pass images representing each of the concepts in a prompt. However, as T2I models are capable of generating many concepts successfully, an efficient strategy would be passing only concepts they struggle to generate. To identify challenging concepts, we generate an initial image with a T2I model. Then, we query a VLM with the initial prompt and image, asking it to decide if they match. If not, we guide the VLM to generate image captions for retrieval, by asking it to identify missing concepts in the image, focusing on content and style, since these are easy to convey through visual cues. We use an example so it knows to only return short generic concepts as an answer. These concepts are the ones we should retrieve and use as references. However, as demonstrated in Tab. 3, image retrieval from brief, generic concept descriptions yields worse results than retrieval from detailed image captions. Therefore, after identifying the missing concepts, we query the VLM to generate detailed image captions for images describing each of the identified missing concepts, which will be used to retrieve exemplar images. By constraining the CoT answers to concise answers, we avoid overthinking and achieve accurate retrievals.

Beyond performance gains on rare and fine-grained concepts, the guided CoT adds interpretability, as we can inspect which concepts were identified as missing. The approach does rely on the diagnostic accuracy of the VLM; therefore, we performed robustness experiments over various VLMs and explored which ones could be trusted for this task (see Appendix E.5 for more details). All the used prompts can be found in Appendix C.

## 3.2 RETRIEVE AND USE REFERENCE IMAGES

We aim to retrieve images that could be described by the generated image captions from an external dataset. To retrieve images matching a given caption, we compare the caption to all the images in the retrieval-dataset using a text-image similarity metric and retrieve the most similar images. Text-to-image retrieval is an active research field (Radford et al., 2021; Zhai et al., 2023; Ray et al., 2024; Vendrow et al., 2024), where no single method is perfect. Retrieval is especially hard when the dataset does not contain an exact match to the query (Biswas & Ramnath, 2024) or when the task is fine-grained retrieval, which depends on subtle details (Wei et al., 2022b). Therefore, we experimented with multiple retrieval strategies (see Tab. 4). First, we tried cosine similarity between CLIP (Radford et al., 2021) and SigLIP (Zhai et al., 2023) text and image embeddings. Next, following a common retrieval workflow, in which first image candidates are being retrieved using pre-computed embeddings, and then the retrieved candidates are being re-ranked using a different, often more expensive but accurate, method (Vendrow et al., 2024), we additionally experimented with some re-ranking methods of reference candidates. Although re-ranking sometimes yields better results compared to using cosine similarity between CLIP embeddings (see Tab. 4), the difference was not significant in most of our experiments. Therefore, for simplicity, in our experiments we used cosine similarity between CLIP embeddings as our similarity metric. The retrieval of good candidates using CLIP raises a question. Some of the base models, e.g. SDXL (Podell et al., 2024), use CLIP as a text-encoder, so how can they benefit from references retrieved by the same model? Our empirical experiments, exemplified in Fig. 2, show that while SDXL struggles to generate some concepts, they are retrieved successfully using CLIP. We hypothesize that retrieval is an easier task than generation. Hence, even though the model cannot generate some concepts, it can still benefit and learn from images representing them, which can be retrieved using CLIP.

After relevant images are retrieved, we can use image conditioning methods to condition T2I models using our reference images by augmenting the input prompt to contain the retrieved images as examples. Formally, given a prompt $p$, $n$ concepts, and a compatible image for each concept, we augment the prompt with the following template: "According to these examples of $<c_1>$:$<img_1>$, ..., $<c_n>$:$<img_n>$, generate $<p>$", where $c_i$ is a compatible image caption of the image $<img_i>$, for $i \in [1, n]$. This prompt allows models to learn missing concepts from the images, guiding them to generate the required result. We experimented with three models and conditioning methods (see Sec. 4), demonstrating our method is model- and conditioning method-agnostic.

**Validation options:** The design of our method allows using a threshold as assurance of reference quality. This way, when retrieving a reference, a threshold could be applied, so only good references, i.e., ones that pass the threshold, would be used. Additionally, we can optionally iterate the generation loop until the VLM indicates alignment between the text prompt and the generated image.

## 4 EXPERIMENTS

To evaluate the effectiveness of our method, we performed automatic and human evaluations. To assess its adaptability to different models, we experimented with applying *ImageRAG* to Omni-Gen (Xiao et al., 2024), SDXL (Podell et al., 2024) through IP-Adapter (Ye et al., 2023), and FLUX (Labs, 2023) through OminiControl (Tan et al., 2024). In most experiments, we used GPT as our VLM and CLIP as our retrieval method, as they performed best overall, and a 350K images subset of LAION (Schuhmann et al., 2022) as our retrieval-dataset for generic usage. We additionally tested the abilities of different VLMs to identify rare concepts (see Appendix E.5) and experimented with different retrieval strategies (see Tab. 4), and with specialized datasets instead of a generic one (see Appendix E.3). Appendix A contains additional implementation details.

Table 1: GPTScore comparisons of fine-grained image generation with text-to-image models. First-part columns feature OmniGen-based models, middle-part columns feature FLUX-based models, and last-part columns feature SDXL-based models. In each part, best results are **bolded**.

| Dataset | OmniGen | GraPE-O | ImageRAG-O | FLUX | ImageRAG-F | SDXL | ImageRAG-SD |
|---|---|---|---|---|---|---|---|
| ImageNet | 0.75 | 0.68 | **0.88** | 0.84 | **0.9** | 0.86 | **0.92** |
| iNaturalist | 0.07 | 0.06 | **0.56** | 0.07 | **0.31** | 0.51 | **0.70** |
| CUB | 0.57 | 0.45 | **0.73** | 0.79 | **0.85** | 0.94 | **0.97** |

Table 2: Comparisons on fine-grained image generation with T2I models. For each set, we report mean ($\pm$ standard error) CLIP, SigLIP text-to-image similarities, and DINO feature similarity between real and generated images. Second-part rows feature OmniGen-based models, third-part feature FLUX-based models, and bottom feature SDXL-based models. In each part, best results are **bolded**.

| | ImageNet | | | iNaturalist | | | CUB | | |
|---|---|---|---|---|---|---|---|---|---|
| | CLIP ↑ | SigLIP ↑ | DINO ↑ | CLIP ↑ | SigLIP ↑ | DINO ↑ | CLIP ↑ | SigLIP ↑ | DINO ↑ |
| Pixart-$\Sigma$ | $0.262^{\pm0.001}$ | $0.121^{\pm0.001}$ | $0.691^{\pm0.003}$ | $0.162^{\pm0.002}$ | $0.027^{\pm0.002}$ | $0.611^{\pm0.002}$ | $0.232^{\pm0.004}$ | $0.101^{\pm0.003}$ | $0.736^{\pm0.004}$ |
| OmniGen | $0.247^{\pm0.002}$ | $0.122^{\pm0.001}$ | $0.692^{\pm0.003}$ | $0.155^{\pm0.002}$ | $0.014^{\pm0.001}$ | $0.595^{\pm0.002}$ | $0.231^{\pm0.005}$ | $0.109^{\pm0.005}$ | $0.747^{\pm0.005}$ |
| GraPE-O | $0.251^{\pm0.002}$ | $0.123^{\pm0.001}$ | $0.692^{\pm0.003}$ | $0.157^{\pm0.002}$ | $0.016^{\pm0.002}$ | $0.604^{\pm0.001}$ | $0.240^{\pm0.005}$ | $0.115^{\pm0.003}$ | $0.747^{\pm0.005}$ |
| ImageRAG-O | $\mathbf{0.264}^{\pm0.001}$ | $\mathbf{0.134}^{\pm0.001}$ | $\mathbf{0.708}^{\pm0.002}$ | $\mathbf{0.197}^{\pm0.002}$ | $\mathbf{0.095}^{\pm0.002}$ | $\mathbf{0.701}^{\pm0.002}$ | $\mathbf{0.253}^{\pm0.003}$ | $\mathbf{0.125}^{\pm0.002}$ | $\mathbf{0.760}^{\pm0.003}$ |
| FLUX | $0.271^{\pm0.001}$ | $0.137^{\pm0.001}$ | $0.698^{\pm0.002}$ | $0.222^{\pm0.002}$ | $0.065^{\pm0.002}$ | $0.654^{\pm0.002}$ | $0.267^{\pm0.003}$ | $0.135^{\pm0.002}$ | $0.746^{\pm0.004}$ |
| ImageRAG-F | $\mathbf{0.277}^{\pm0.001}$ | $\mathbf{0.140}^{\pm0.001}$ | $\mathbf{0.705}^{\pm0.002}$ | $\mathbf{0.238}^{\pm0.001}$ | $\mathbf{0.083}^{\pm0.002}$ | $\mathbf{0.691}^{\pm0.002}$ | $\mathbf{0.277}^{\pm0.002}$ | $\mathbf{0.144}^{\pm0.002}$ | $\mathbf{0.767}^{\pm0.003}$ |
| SDXL | $0.267^{\pm0.002}$ | $0.136^{\pm0.001}$ | $0.700^{\pm0.003}$ | $\mathbf{0.259}^{\pm0.002}$ | $0.096^{\pm0.002}$ | $0.698^{\pm0.003}$ | $\mathbf{0.315}^{\pm0.001}$ | $0.172^{\pm0.003}$ | $0.782^{\pm0.002}$ |
| ImageRAG-SD | $\mathbf{0.274}^{\pm0.001}$ | $\mathbf{0.141}^{\pm0.001}$ | $\mathbf{0.709}^{\pm0.002}$ | $0.243^{\pm0.002}$ | $\mathbf{0.118}^{\pm0.001}$ | $\mathbf{0.724}^{\pm0.002}$ | $0.314^{\pm0.001}$ | $\mathbf{0.174}^{\pm0.002}$ | $\mathbf{0.784}^{\pm0.001}$ |

## 4.1 Quantitative comparisons

We evaluate the ability of *ImageRAG* to improve T2I generation of rare and fine-grained concepts by comparing the results of different base models with their results when applying *ImageRAG* to them. As additional baselines, we compare with Pixart-$\Sigma$ (Chen et al., 2025), and the OmniGen-based version of GraPE (Goswami et al., 2024). The last is an iterative LLM-based image generation method which employs editing tools to insert missing objects. Following rare concept generation works (Samuel et al., 2024b; Pan et al., 2025), we use the fine-grained datasets ImageNet (Deng et al., 2009), iNaturalist (Van Horn et al., 2018), and CUB (Wah et al., 2011) for evaluation. Samuel et al. (2024b) reported that 25% of ImageNet classes are in the tail of LAION (Schuhmann et al., 2022), and most of the classes in CUB and iNaturalist are in its tail. For iNaturalist, we use the first 1000 classes. Additional experimental results over the Flowers (Nilsback & Zisserman, 2008), Dogs (Khosla et al., 2011), and Cars (Krause et al., 2013) datasets are reported in Appendix E.1. Following prior work (Pang et al., 2024; Ruiz et al., 2023; Zhang et al., 2023), we evaluate all methods with the commonly used similarity metrics CLIP (Radford et al., 2021) and DINO (Zhang et al., 2022), in Tab. 2. For fairness, we use open-CLIP for evaluation, and OpenAI CLIP for retrieval. We additionally report SigLIP (Zhai et al., 2023) embedding similarity, which outperforms CLIP on several classification and retrieval tasks. When examining these scores alone, improvement seems mild. However, these metrics gauge only coarse semantic similarity: for instance, two bird species appear similar in the embedding space despite possessing meaningful distinctions required for concept-specific generation. Therefore, we additionally report GPTScores (Peng et al., 2025), which has been shown to correlate with human judgment for personalized image generation evaluation, where concepts are inherently unknown to the model. Moreover, we conduct user studies (see Sec. 4.1.1) and supply visual examples (Fig. 5 and Appendix B), where improvements are much clearer, as they capture the intended evaluation signal more faithfully. As Tabs. 1 and 2 demonstrate, all base models results improve when using *ImageRAG* for rare concept and fine-grained generation.

### 4.1.1 User studies

To evaluate our results further, we conducted a user study with 67 participants, including two types of comparative studies (977 comparisons) and an absolute study (231 answers).

**Comparative studies:** 1) comparing OmniGen, SDXL, and FLUX with and without our method, and 2) comparing our results to retrieval-based models trained for the task of image generation using retrieved images: RDM (Blattmann et al., 2022), knn-diffusion (Sheynin et al., 2022), and

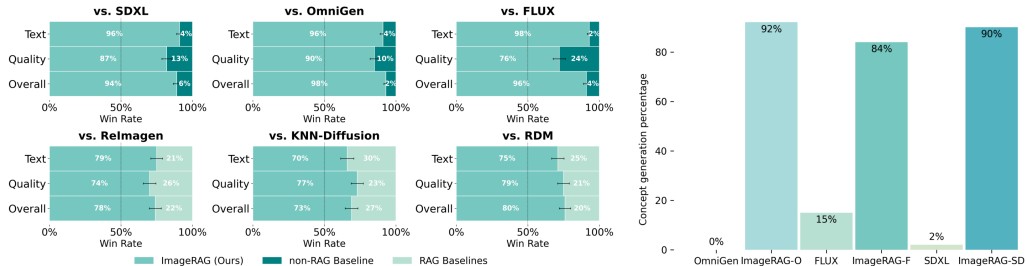

Figure 4: **User study results.** Left: Users preference percentage of our method compared to other methods in terms of text alignment, visual quality, and overall preference. Right: Percentage of rare concept generation per model by users' rating of whether concepts are contained in generated images.

Table 3: Ablation studies over OmniGen. "Rephrased prompt": only prompt rephrasing without image references. "Retrieve concepts": using the missing concepts for retrieval instead of using detailed image captions, "Retrieve prompt": using the prompt for retrieval. Best results are **bolded**.

| | ImageNet | | | CUB | | |
|---|---|---|---|---|---|---|
| | CLIP ↑ | SigLIP ↑ | DINO ↑ | CLIP ↑ | SigLIP ↑ | DINO ↑ |
| OmniGen | $0.247 \pm 0.002$ | $0.122 \pm 0.001$ | $0.692 \pm 0.003$ | $0.231 \pm 0.005$ | $0.109 \pm 0.003$ | $0.747 \pm 0.005$ |
| Rephrased prompt-O | $0.248 \pm 0.002$ | $0.124 \pm 0.042$ | $0.696 \pm 0.003$ | $0.230 \pm 0.005$ | $0.107 \pm 0.004$ | $0.750 \pm 0.005$ |
| Retrieve concepts-O | $0.258 \pm 0.002$ | $0.130 \pm 0.001$ | $0.694 \pm 0.003$ | $0.240 \pm 0.004$ | $0.113 \pm 0.003$ | $0.719 \pm 0.006$ |
| Retrieve prompt-O | $0.258 \pm 0.002$ | $0.130 \pm 0.001$ | $0.691 \pm 0.003$ | $0.246 \pm 0.004$ | $0.120 \pm 0.003$ | $0.736 \pm 0.005$ |
| ImageRAG-O | $\mathbf{0.264 \pm 0.001}$ | $\mathbf{0.134 \pm 0.001}$ | $\mathbf{0.708 \pm 0.002}$ | $\mathbf{0.253 \pm 0.003}$ | $\mathbf{0.125 \pm 0.002}$ | $\mathbf{0.760 \pm 0.003}$ |

ReImagen (Chen et al., 2022). Since these are largely proprietary models with no API, we compared to images and prompts published in their papers. In each comparison, participants chose between an *ImageRAG* result and a baseline image based on text alignment, visual quality, and overall preference. Since some prompts contain uncommon concepts, we supply a real image of the least familiar concept in each prompt (not taken from our dataset). As demonstrated in Fig. 4 (left), participants favored *ImageRAG* over all other methods in all three criteria of text alignment, visual quality, and overall preference.

**Absolute study:** to assert our method not only improves the current results but also generates the rare concepts, we performed an absolute study. We asked participants which of the results of all baseline models with and without our method contain a reference object. Note: we only used images where our method ran, meaning where the VLM decided the initial images did not match the prompt. As shown in Fig. 4, indeed our results contain the reference object in most cases; 92% (OmniGen), 90% (SDXL), and 84% (FLUX), while the baseline models originally did not contain it in most cases (indicating that the VLM was able to identify the missing concepts accurately). Appendix F supplies more information about the studies, including questions and visual examples of comparisons presented in it for each retrieval-based generation model (Fig. 6), OmniGen (Fig. 7), SDXL (Fig. 8), and FLUX (Fig. 9), with and without *ImageRAG*.

## 4.2 QUALITATIVE EXAMPLES

Fig. 5 shows rare concept generation examples by OmniGen, FLUX, and SDXL, with and without ImageRAG. In all examples, the base models did not generate the required concepts and guessed an animal based on the prompt. Moreover, even if they were able to deduce the required animal from the context, e.g., a bird in the left-most prompt, they did not generate the exact bird species requested. When supplied with relevant references using *ImageRAG*, all methods succeeded in the generation tasks. Fig. 10 presents additional visual results with more complex and creative prompts, and Fig. 14 shows personalized generation examples with rare concepts. In Appendix E.6 we discuss diversity and present diverse generation results using each model with different seeds.

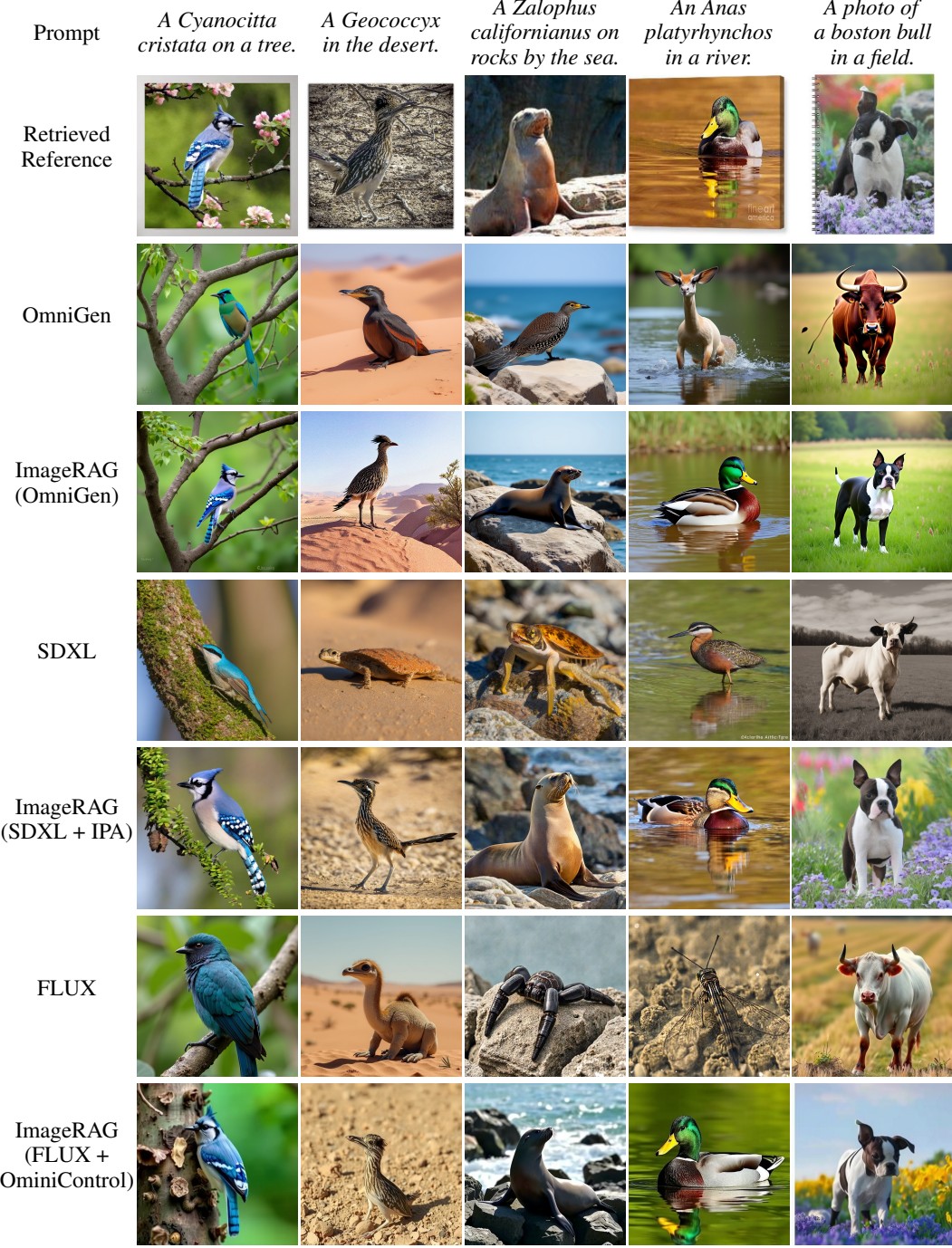

Figure 5: **Qualitative comparisons: rare concept generation.** Examples from ImageNet, CUB, and iNaturalist. The top image column is the retrieved reference using *ImageRAG* for each prompt. OmniGen, SDXL, and FLUX all struggle to generate the uncommon concepts, however when using *ImageRAG*, all models successfully generate the required prompts.

Table 4: Similarity metric ablation study over OmniGen. Results of our method using different similarity metrics for image retrieval. Best results are **bolded**.

| | ImageNet | | | CUB | | |
|---|---|---|---|---|---|---|
| | CLIP ↑ | SigLIP ↑ | DINO ↑ | CLIP ↑ | SigLIP ↑ | DINO ↑ |
| GPT Re-rank | **0.265 ± 0.001** | **0.135 ± 0.001** | 0.707 ± 0.002 | **0.255 ± 0.004** | **0.125 ± 0.003** | 0.762 ± 0.004 |
| BM25 Re-rank | 0.264 ± 0.001 | 0.134 ± 0.001 | 0.707 ± 0.002 | 0.253 ± 0.003 | 0.123 ± 0.003 | **0.763 ± 0.004** |
| SigLIP | 0.259 ± 0.006 | 0.133 ± 0.001 | 0.704 ± 0.002 | 0.243 ± 0.004 | 0.116 ± 0.003 | 0.761 ± 0.004 |
| CLIP | 0.264 ± 0.001 | 0.134 ± 0.001 | **0.708 ± 0.002** | 0.253 ± 0.003 | **0.125 ± 0.002** | 0.760 ± 0.003 |

## 4.3 ABLATION STUDIES

To evaluate the contribution of each part of our method, we conducted ablation studies and reported our results in Tabs. 3 and 8. First, to ensure the performance gap is not based on an LLM interpreting rare words, we evaluated the base models over rephrased text prompts, without providing reference images. To do so, we asked GPT to rephrase the prompts, to make them easier for a T2I generative model, explicitly asking it to change rare words to their description if necessary. Full prompt can be found in Appendix C. As shown, rephrasing was not enough for a meaningful improvement in the results ("Rephrased prompt" in Tabs. 3 and 8). Next, we investigate the importance of using detailed image captions for retrieval, rather than using the original prompt or the missing concepts. We do so by evaluating *ImageRAG* when retrieving images similar to the prompt directly ("Retrieve prompt" in the tables), as done by previous works (Blattmann et al., 2022; Sheynin et al., 2022; Chen et al., 2022), or images similar to missing concepts, without generating compatible image captions for each concept ("Retrieve concepts" in Tabs. 3 and 8). While retrieval with each of them improves the initial results, retrieving detailed captions improves the results even further. Fig. 15 presents examples.

We additionally investigate the effect of different similarity metrics for retrieval (Tab. 4). We tried CLIP (Radford et al., 2021), SigLIP (Zhai et al., 2023), re-ranking of retrieved candidates by BM25 (Robertson et al., 2009) over image captions generated by GPT, and by GPT (Hurst et al., 2024). Re-ranking was performed over the top 3 candidates retrieved from each of CLIP and SigLIP. Although re-ranking with GPT produced slightly better results, the improvement was not significant enough to justify the cost of applying this complex strategy vs. a more straightforward CLIP metric. Hence, our other experiments use CLIP. Nevertheless, all retrieval strategies improved the generation abilities of the base model by providing helpful references. Next, as we rely on a VLM to identify rare concepts in images, we performed a VLM robustness experiment explained in Appendix E.5 with various VLMs. GPT performed best and can successfully identify rare concepts, hence we can rely on it for our method. However, Gemini and Qwen also performed well, and could be potentially used instead of GPT. We further investigate the effect of the retrieval-dataset size (discussed in Appendix E.4). Typically, the larger the dataset, the better the results. However, even a relatively small dataset can improve results. Moreover, we experiment with using a specific proprietary retrieval dataset and show that the more relevant the dataset is, the better the results are (Appendix E.3).

## 5 LIMITATIONS

The capabilities of *ImageRAG* depend on the retrieval data and method, and on the base model. **Retrieval data:** our ability to aid generation depends on the retrieval dataset. E.g., our method will not help when generating a specific dog breed from a dataset of birds, as in Fig. 12, left. Specifically, we noticed that if the retrieved image is completely unrelated to the text (as in a bird and a dog), the models tend to rely almost exclusively on the text. This is likely because semantically unrelated concepts do not attend to each other. As presented in Tab. 6, the more relevant the retrieval dataset is, the more accurate the generation will be.

**Retrieval method:** performance is tied to the quality of the retriever. CLIP-based retrieval inherits its weaknesses, such as poor counting (Paiss et al., 2023). Additionally, we rely on the used VLM to decide whether we should apply our method and identify gaps. Although VLMs are powerful and usually answer correctly, even for rare concepts, as evaluated in Appendix E.5, sometimes they misidentify an initial image as a match to the original prompt, leading to not applying our method. This happens especially if the prompt could be interpreted in multiple ways. For example, for the

prompt "A photo of a love in the mist", OmniGen generated an initial image of a couple in the mist. This does match the prompt, hence the VLM identified it as correct; however, our intention was the flower "love in the mist". In that case, we can either add "the flower" to the prompt or explicitly indicate the mismatch in the query so the VLM knows it should refer to the other meaning.

**Underlying model:** some concepts remain difficult for the generator itself; e.g., both OmniGen and SDXL struggle to reproduce text even when provided with text references. Fig. 12 shows visual limitations examples.

## 6    CONCLUSION

In this work, we propose a simple yet effective approach for applying RAG to pretrained T2I models. We demonstrate that incorporating relevant image references improves the rare concept generation abilities of T2I models. By leveraging a VLM, we enable dynamic retrieval of suitable reference images. Our experiments span three distinct models, illustrating the adaptability of our method across different model types. Our findings highlight that image references can enhance image generation with minimal modifications to existing models, thereby broadening their practical applicability.

## 7    ETHICS STATEMENT

The development of *ImageRAG* introduces enhancement possibilities for image generation models, enabling rare or fine-grained concept generation. While these advancements hold promise for creative industries, personalized content creation, and visualization, they also raise ethical concerns, including potential misuse for harmful content such as deepfakes and the use of private data. Therefore, transparency in data usage and adherence to privacy regulations are essential. We condemn any misuse of the proposed technology, and actively research tools to identify and prevent malicious usage of generative models. Moreover, unlike pre-trained models, which encode all knowledge in their weights, our method uses an external dataset. This allows the users to filter out unwanted images. This way, a possible solution to avoid copyright issues is to only use images with appropriate licenses in the retrieval dataset, while following relevant restrictions if they apply. In the retrieval scenario, it is also easier to directly attribute the source of the image. This should allow users to, e.g., credit the original creators, or ensure new images follow any license restrictions required by the publishers of the original image.

## 8    REPRODUCIBILITY STATEMENT

We provide detailed implementation details in Sec. 4 and Appendix A, including retrieval-dataset details, models used, hyperparameters, etc. The used prompts are also provided in Appendix C.

To further facilitate reproducibility, we will release full code for applying ImageRAG over FLUX, SDXL, and OmniGen upon publication.

## ACKNOWLEDGMENTS

We thank Gal Chechik and Daniel Arkushin for helpful discussions. This research was partially funded by ISF grant numbers 1337/22, 1574/21.

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

## A    IMPLEMENTATION DETAILS

We use a random subset of LAION (Schuhmann et al., 2022) containing 350K images as the dataset from which we retrieve images. As a retrieval similarity metric, we use CLIP "ViT-B/32". We do not use a retrieval threshold; however, we notice that all retrieved images in our experiments have a similarity greater than 0.26. For a VLM we use GPT-4o-2024-08-06 (Hurst et al., 2024) with a temperature of 0 for higher consistency (unless GPT fails to find concepts, see Appendix D). Full GPT prompts are supplied in Appendix C. As our T2I generation base models we use SDXL (Podell et al., 2024) with the ViT-H IP-adapter (Ye et al., 2023) plus version ("ip-adapter-plus_sdxl_vit-h"), using 0.5 for the ip_adapter_scale, FLUX (Labs, 2023) schnell with OminiControl (Tan et al., 2024), using 50 steps, and OmniGen (Xiao et al., 2024) with the default parameters (2.5 guidance scale, 1.6 image guidance scale). As OmniGen only supports 3 images as context, we use up to $n = 3$ concepts for each prompt and 1 image per concept. For SDXL and FLUX, as the encoders we used are limited to 1 image, we use 1 concept and 1 image.

The time and resources required to run our method depend on the used baseline model and the used retrieval-dataset. However, CLIP embeddings for the retrieval-dataset are pre-computed once, so at inference finding compatible reference images operates in sub-second time. Our experiments show that adding reference images to the base generation models does not meaningfully increase the latency of the models. Therefore, the only additional time our method requires is the time required to perform the iterative chain-of-thought with the VLM through 3 API calls with an image, for the decision, identification, and image caption-generation parts, and generating the second image, which depends on the base generative model and could be optimized per model, as described in Sec. 3. With GPT-4o and Qwen-2.5, the CoT process takes 10-30 seconds in total, where for GPT it depends on network latency and API queue delays. If using GPT, the cost of the 3 calls together is approximately 0.01-0.05$.

## B    VISUAL EXAMPLES

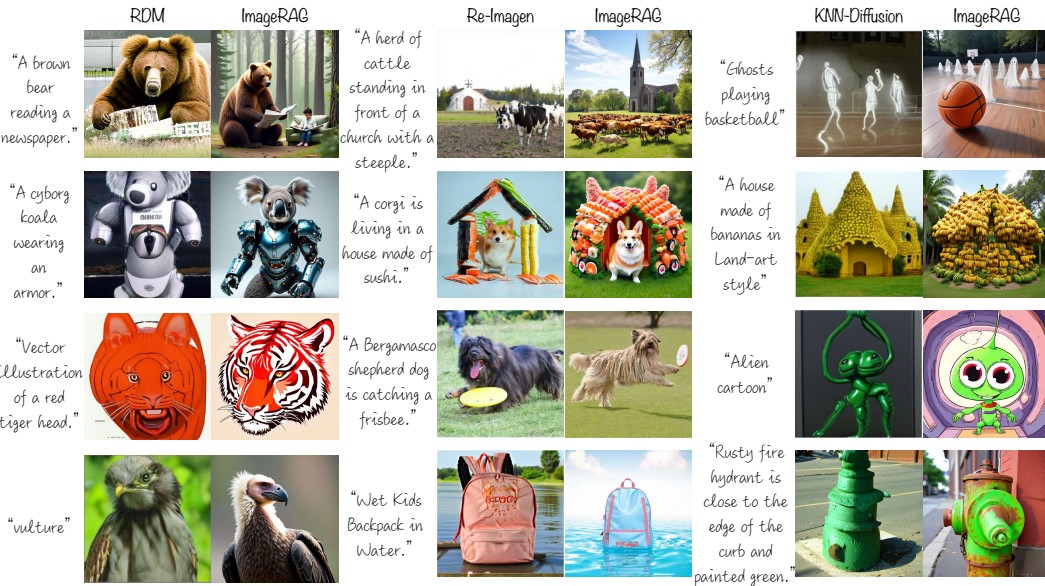

Figure 6: **Comparisons between ImageRAG and different methods using retrieval for generation.** Prompts and results of all other methods are taken from their papers. The methods we compared to are RDM Blattmann et al. (2022), Re-Imagen Chen et al. (2022), and KNN-Diffusion Sheynin et al. (2022).

Pair examples of rare and fine-grained concept generation with and without *ImageRAG* are presented in Fig. 7 (OmniGen examples), Fig. 8 (SDXL examples), and Fig. 9 (FLUX examples).

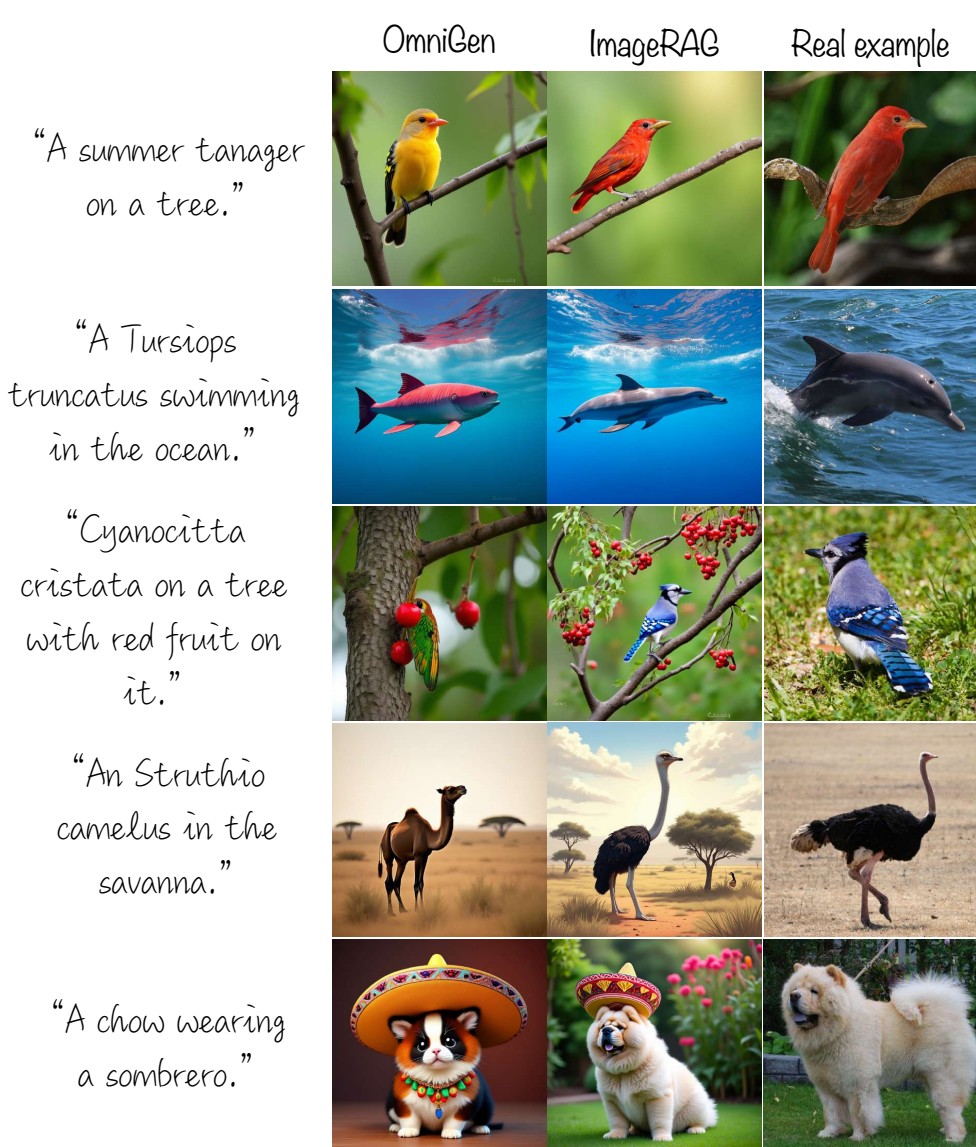

Figure 7: **Examples of rare concept generation using ImageRAG with OmniGen.**

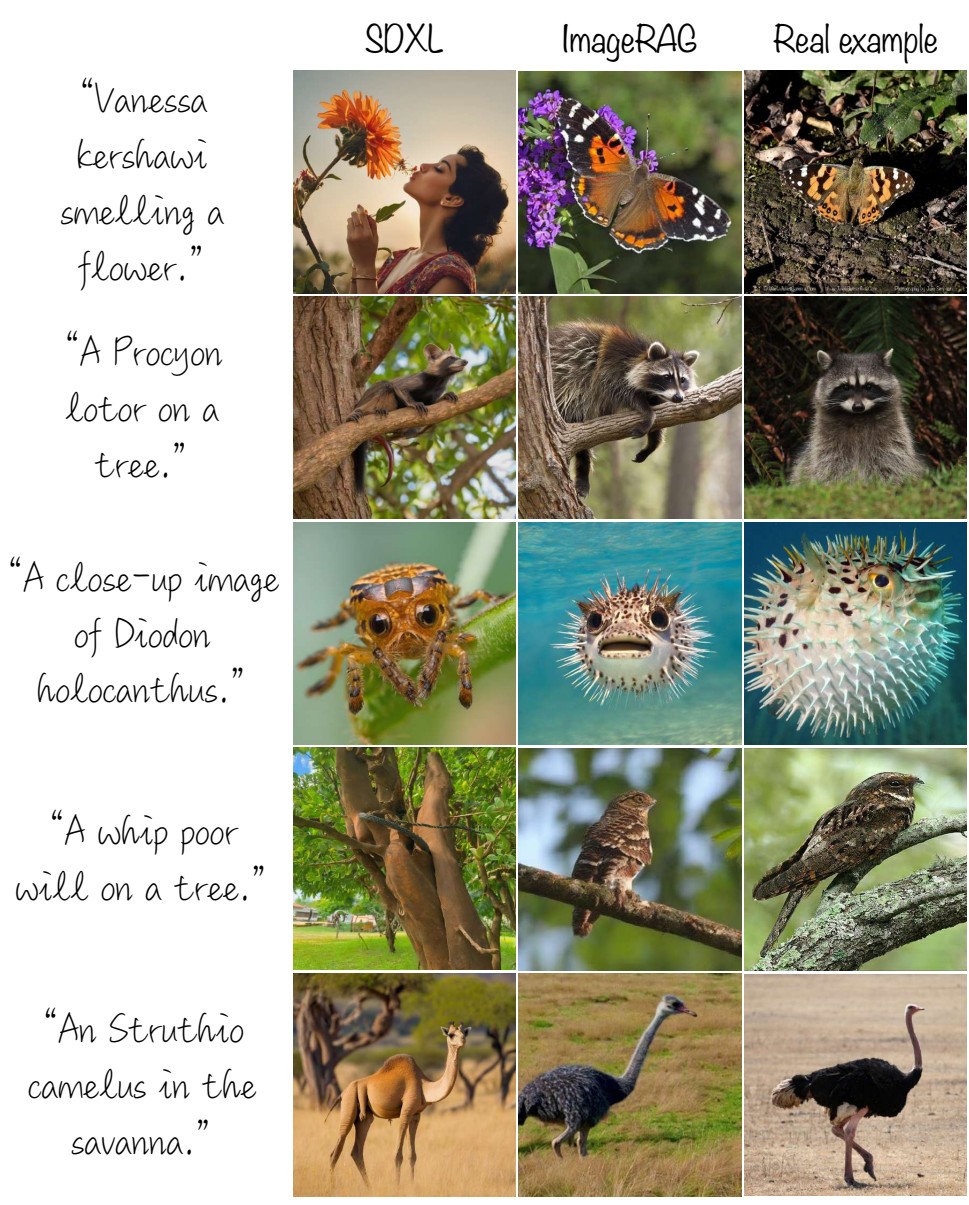

Figure 8: **Examples of rare concept generation using ImageRAG with SDXL.**

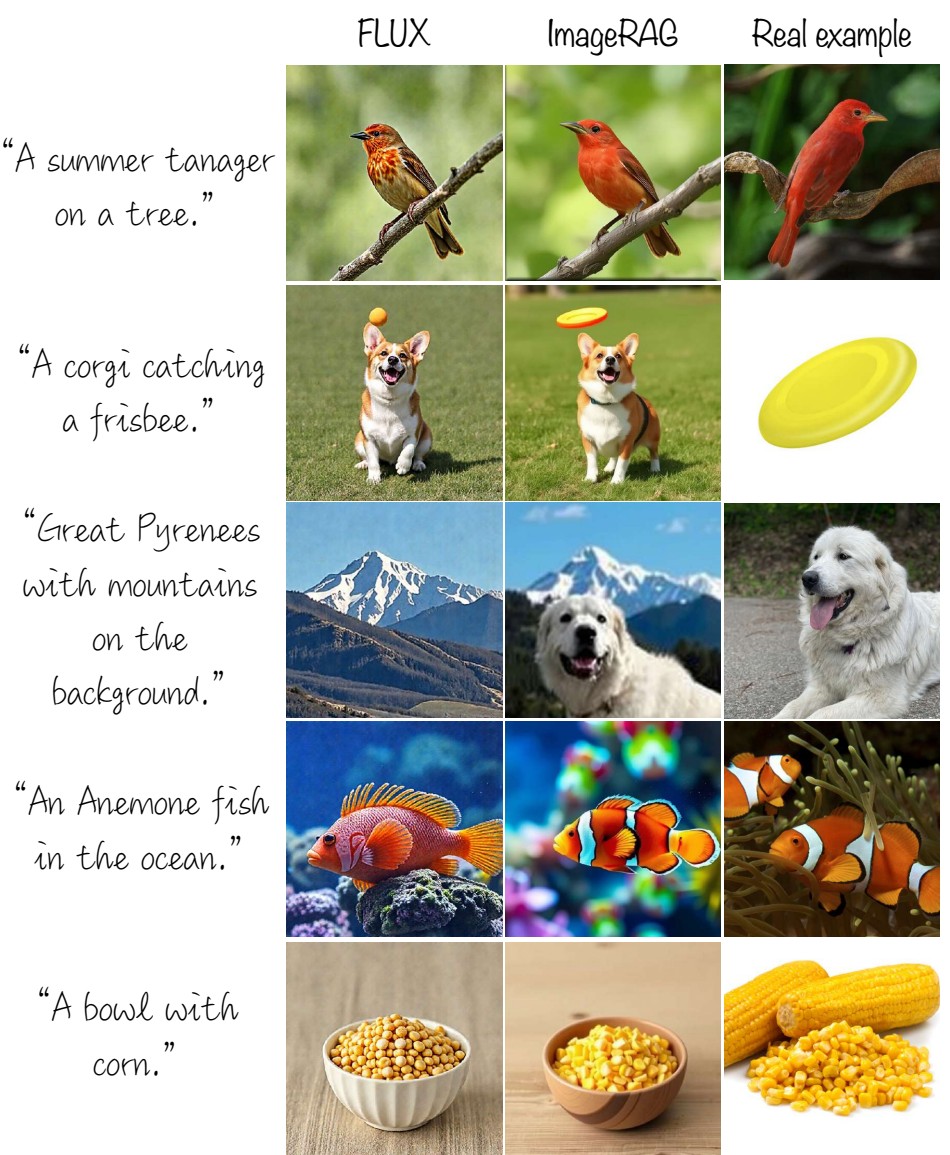

Figure 9: **Examples of rare concept generation using ImageRAG with FLUX.**

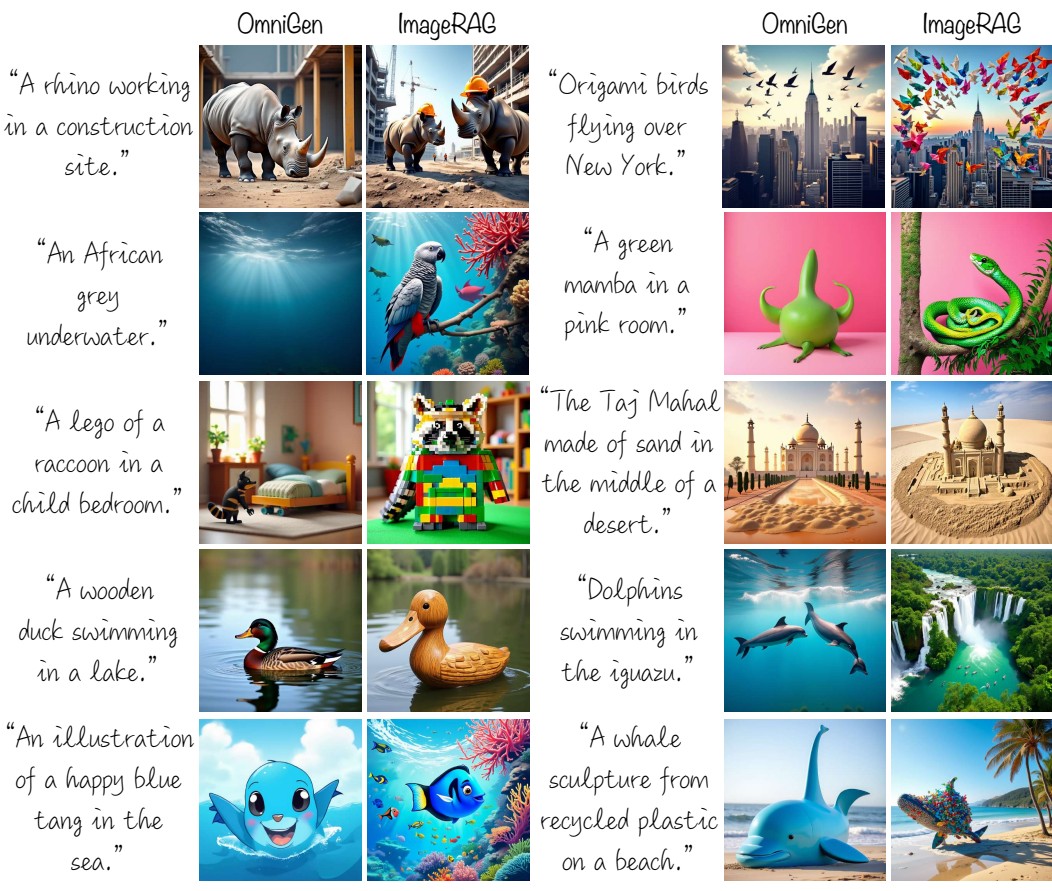

Figure 10: **More creative generation examples.**

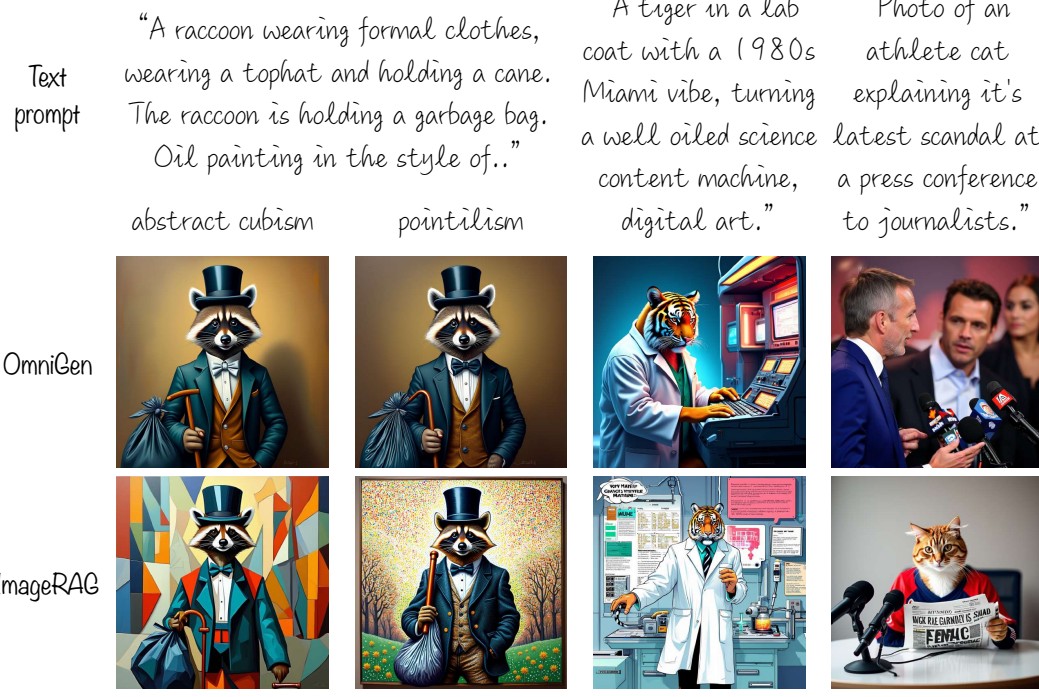

Figure 11: **More long and complex generation examples.**

More creative examples are presented in Fig. 10. Longer and more complex results are presented in Fig. 11.

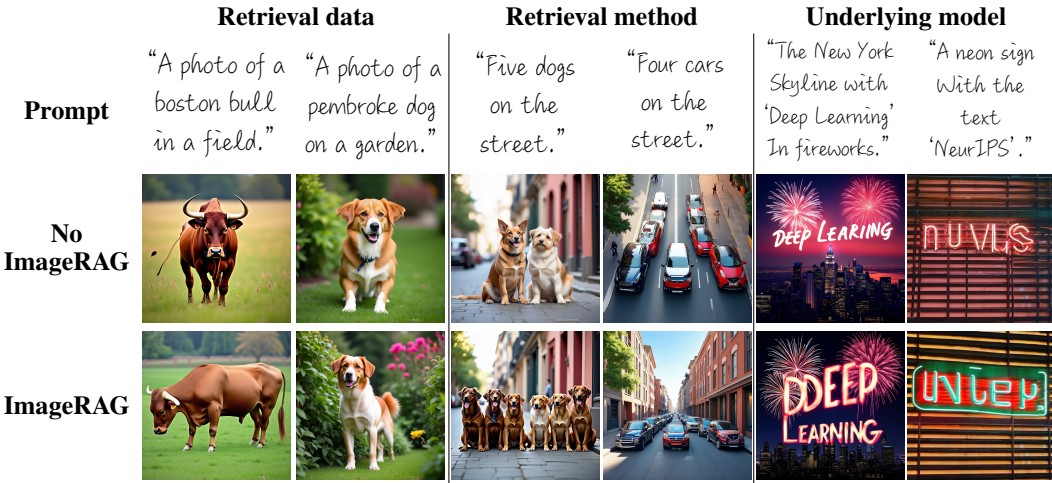

Figure 12: **Limitations of our method.** Left: retrieval data. If the retrieval data does not contain relevant examples, our method cannot help. In the examples above we used CUB as the retrieval dataset, which only contains images of birds, thus it does not help for generation of other concepts such as dog breeds. Middle: retrieval method. Our method relies on the quality of the retrieval method. e.g. when using CLIP — we cannot help with counting Paiss et al. (2023). Right: underlying model. Some concepts are not well learned from images by the base models, such as text. In these cases, our ability to help is limited.

Visual examples of limitations of our method are presented in Fig. 12.

## C  RETRIEVAL-CAPTION GENERATION PROMPTS

Full prompts used for querying GPT in the retrieval-caption generation part of our method:

**Decision:** 'Does this image match the prompt "{prompt}"? Consider both content and style aspects. Only answer yes or no.'

**Missing Concepts Identification:** 'What are the differences between this image and the required prompt? In your answer only provide missing concepts in terms of content and style, each in a separate line. For example, if the prompt is "An oil painting of a sheep and a car" and the image is a painting of a car but not an oil painting, the missing concepts will be:
oil painting style
a sheep'

**Caption Generation:** 'For each concept you suggested above, please suggest an image caption describing an image that explains this concept only. The captions should be stand-alone description of the images, assuming no knowledge of the given images and prompt, that I can use to lookup images with automatically. In your answer only provide the image captions, each in a new line with nothing else other than the caption.'

**Rephrase request prompt:** prompt used for the rephrasing ablation experiment: 'Please rephrase the following prompt to make it easier and clearer for the text-to-image generation model that generated the above image for this prompt. The goal is to generate an image that matches the given text prompt. If the prompt is already clear, return it as it is. Simplify and shorten long descriptions of known objects/entities but DO NOT change the original meaning of the text prompt. If the prompt contains rare words, change those words to a description of their meaning. In your answer only provide the prompt and nothing else. The prompt to be rephrased: "{prompt}".'

Table 5: Comparisons on fine-grained image generation with T2I models. For each set, we report CLIP-T and CLIP-I similarity scores. First-part rows feature OmniGen-based models, second-part feature FLUX-based models, and bottom feature SDXL-based models. In each part, best results are **bolded**.

| | Dogs | | Flowers | | Cars | |
|---|---|---|---|---|---|---|
| | CLIP-T ↑ | CLIP-I ↑ | CLIP-T ↑ | CLIP-I ↑ | CLIP-T ↑ | CLIP-I ↑ |
| OmniGen | 0.27 | 0.52 | 0.28 | 0.58 | 0.24 | 0.47 |
| ImageRAG-O | **0.30** | **0.57** | **0.31** | **0.66** | **0.29** | **0.55** |
| FLUX | 0.30 | 0.62 | 0.30 | 0.66 | 0.31 | 0.60 |
| ImageRAG-F | **0.31** | **0.63** | **0.31** | **0.68** | 0.31 | 0.60 |
| SDXL | 0.34 | 0.61 | 0.34 | 0.68 | 0.34 | 0.62 |
| ImageRAG-SD | 0.34 | **0.62** | 0.34 | **0.69** | 0.34 | 0.62 |

## D  VLM Error Handling

The VLM may sometimes fail to identify the missing concepts in an image, and will respond that it is "unable to respond". In these rare cases, we allow up to 3 query repetitions, while increasing the query temperature in each repetition. Increasing the temperature allows for more diverse responses by encouraging the model to sample less probable words. In most cases, using our suggested step-by-step method yields better results than retrieving images directly from the given prompt (see Tabs. 3 and 8). However, if the VLM still fails to identify the missing concepts after multiple attempts, we fall back to retrieving images directly from the prompt, as it usually means the VLM does not know what is the meaning of the prompt.

## E  Additional Experiments

### E.1  Additional datasets

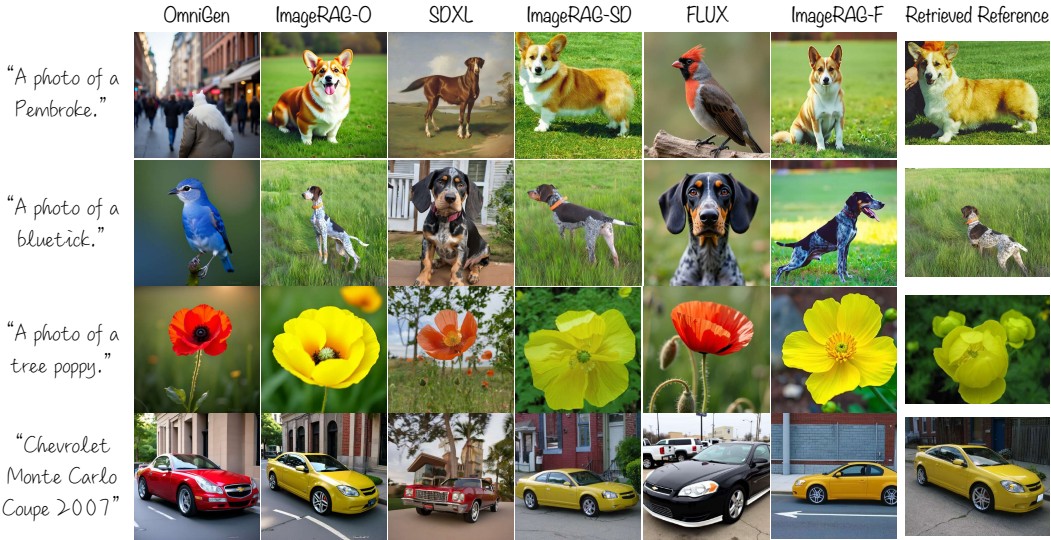

Figure 13: **Visual examples from the Dogs, Flowers, and Cars datasets.**

Following RealRAG (Lyu et al., 2025) we evaluate our method over three additional datasets: Oxford Flowers (Nilsback & Zisserman, 2008), Stanford Dogs (Khosla et al., 2011), and Stanford Cars (Krause et al., 2013) with the metrics CLIP-T (text-image similarity), and CLIP-I (image-image similarity, between generated and ground-truth images). Similarly to RealRAG, we included the dogs,

flowers, and cars datasets in the retrieval dataset, as in our "proprietary data generation" experiment (Appendix E.3). Results are presented in Tab. 5. Visual examples from generations of these sets are presented in Fig. 13.

## E.2 PERSONALIZED GENERATION

For models that support multiple input images, we can combine our method with personalized generation, to generate rare concept combinations with personal concepts. In this case, we use one image for personal content, and 1+ other reference images for missing concepts. For example, given an image of a specific cat, we can generate diverse images of it, ranging from a mug featuring the cat to a lego of it or atypical situations like the cat writing code or teaching a classroom of dogs (Fig. 14).

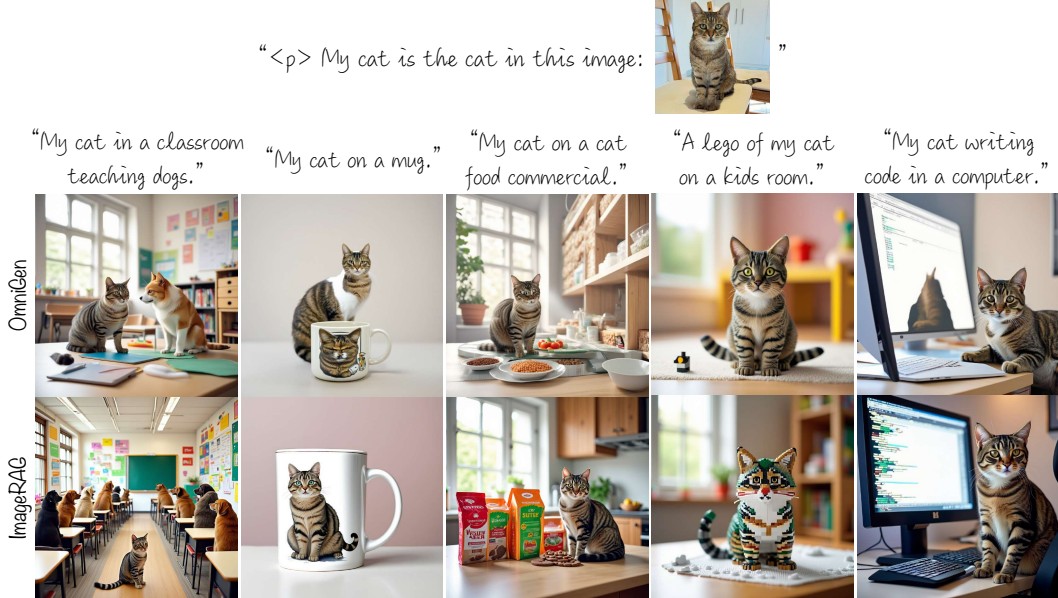

Figure 14: **Personalized generation example.** *ImageRAG* can work in parallel with personalization methods and enhance their capabilities. For example, although OmniGen can generate images of a subject based on an image, it struggles to generate some concepts. Using references retrieved by our method, it can generate the required result.

## E.3 PROPRIETARY DATA GENERATION

Table 6: Proprietary data usage experiment. Results for using each dataset as the retrieval-dataset ("Proprietary-<model>") vs. using our subset from LAION as the retrieval-dataset ("LAION-<model>"). Here, "O" indicates OmniGen based models, "F" indicates FLUX based models, and "SD" indicates SDXL based models. Best results for each model are **bolded**.

|  | ImageNet | | | iNaturalist | | |
|---|---|---|---|---|---|---|
|  | CLIP ↑ | SigLIP ↑ | DINO ↑ | CLIP ↑ | SigLIP ↑ | DINO ↑ |
| LAION-O | 0.264 ± 0.001 | 0.134 ± 0.001 | 0.708 ± 0.002 | 0.197 ± 0.002 | 0.095 ± 0.002 | 0.701 ± 0.002 |
| Proprietary-O | **0.266 ± 0.001** | **0.136 ± 0.001** | **0.710 ± 0.002** | **0.212 ± 0.002** | **0.114 ± 0.001** | **0.732 ± 0.002** |
| LAION-F | 0.271 ± 0.001 | 0.137 ± 0.001 | 0.698 ± 0.002 | 0.222 ± 0.002 | 0.065 ± 0.002 | 0.654 ± 0.002 |
| Proprietary-F | **0.275 ± 0.001** | **0.140 ± 0.001** | **0.703 ± 0.002** | **0.227 ± 0.002** | **0.080 ± 0.002** | **0.682 ± 0.002** |
| LAION-SD | 0.274 ± 0.001 | 0.141 ± 0.001 | 0.709 ± 0.002 | 0.243 ± 0.002 | **0.118 ± 0.001** | 0.724 ± 0.002 |
| Proprietary-SD | **0.288 ± 0.001** | **0.142 ± 0.001** | **0.736 ± 0.003** | **0.251 ± 0.002** | **0.118 ± 0.002** | **0.737 ± 0.002** |

A common use for RAG in NLP is generation based on proprietary data (Lewis et al., 2020), where the retrieval-dataset is proprietary. A similar application in image generation is generating images

based on a proprietary gallery of images; e.g., for personalization, where the gallery is of a personal concept or a company brand, or a private collection of images that could broaden the knowledge of a model. Our LAION-based experiments explored the scenario where a user has access to a general, large-scale set. Here, we further evaluate the performance of *ImageRAG* when we have access to a potentially smaller, specialized dataset. We repeat the experiments with the datasets used in Tab. 2, but this time retrieve samples from within each dataset rather than from the LAION subset. Results are reported in Tabs. 6 and 7. We observe that although applying our method with the generic dataset of a LAION subset already improves the results, they improve even further when using proprietary retrieval-datasets.

## E.4 ADDITIONAL ABLATIONS

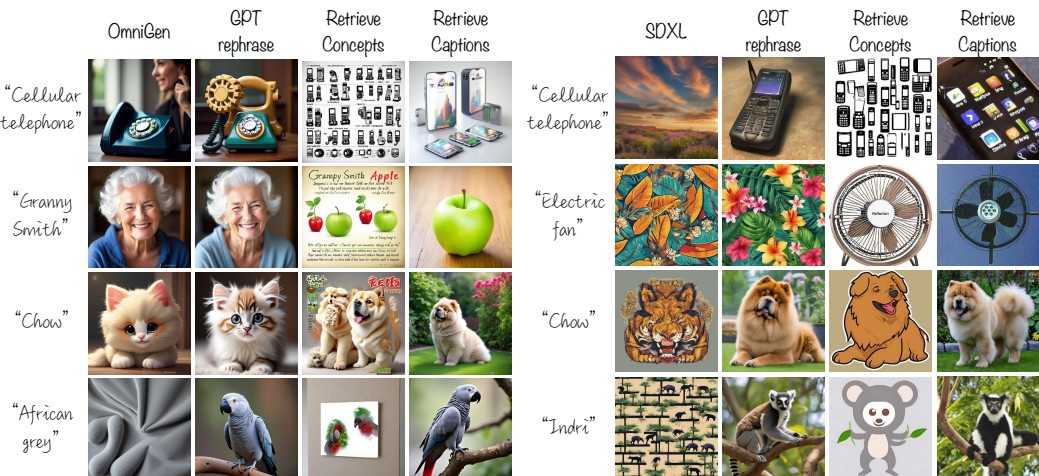

Figure 15: **Visual examples of the ablation studies.** Left: Omnigen, right: SDXL.

Ablations studies over SDXL, as explained in Sec. 4 under ablations, are reported in Tab. 8. Fig. 15 presents visual examples of the ablations over Omnigen and SDXL.

**Retrieval-dataset size:** we investigate the effect of the retrieval-dataset size. We tested our method over ImageNet (Deng et al., 2009) and Aircraft (Maji et al., 2013) when using 1000, 10,000, 100,000, and 350,000 examples from LAION Schuhmann et al. (2022). Fig. 16 shows that increasing the dataset size typically leads to better results. However, even using a relatively small dataset can already lead to improvements. For OmniGen, 1000 examples were enough to see an improvement over the baseline model. SDXL has a stronger baseline, hence more examples were needed for improvement.

Table 7: Additional proprietary data usage experiments. Results for using each dataset as the retrieval-dataset ("Proprietary-<model>") vs. using our subset from LAION as the retrieval-dataset ("LAION-<model>"). Here, "O" indicates OmniGen based models, "SD" indicates SDXL based models. Best results for each model are **bolded**.

| | CUB | | | Aircraft | | |
|---|---|---|---|---|---|---|
| | CLIP ↑ | SigLIP ↑ | DINO ↑ | CLIP ↑ | SigLIP ↑ | DINO ↑ |
| LAION-O | $0.253 \pm 0.003$ | $0.125 \pm 0.002$ | $0.760 \pm 0.003$ | $0.228 \pm 0.006$ | $0.103 \pm 0.005$ | $0.747 \pm 0.010$ |
| Proprietary-O | $\textbf{0.269} \pm \textbf{0.003}$ | $\textbf{0.136} \pm \textbf{0.002}$ | $\textbf{0.773} \pm \textbf{0.004}$ | $\textbf{0.244} \pm \textbf{0.007}$ | $\textbf{0.109} \pm \textbf{0.005}$ | $\textbf{0.786} \pm \textbf{0.010}$ |
| LAION-F | $0.267 \pm 0.003$ | $0.135 \pm 0.002$ | $0.746 \pm 0.004$ | $0.266 \pm 0.006$ | $0.128 \pm 0.005$ | $0.738 \pm 0.002$ |
| Proprietary-F | $\textbf{0.291} \pm \textbf{0.002}$ | $\textbf{0.153} \pm \textbf{0.002}$ | $\textbf{0.770} \pm \textbf{0.002}$ | $\textbf{0.269} \pm \textbf{0.006}$ | $\textbf{0.137} \pm \textbf{0.004}$ | $\textbf{0.753} \pm \textbf{0.087}$ |
| LAION-SD | $\textbf{0.314} \pm \textbf{0.001}$ | $0.174 \pm 0.002$ | $0.784 \pm 0.001$ | $0.272 \pm 0.005$ | $0.141 \pm 0.005$ | $0.756 \pm 0.011$ |
| Proprietary-SD | $\textbf{0.314} \pm \textbf{0.002}$ | $\textbf{0.175} \pm \textbf{0.001}$ | $\textbf{0.786} \pm \textbf{0.003}$ | $\textbf{0.280} \pm \textbf{0.005}$ | $\textbf{0.152} \pm \textbf{0.003}$ | $\textbf{0.785} \pm \textbf{0.009}$ |

Table 8: Ablation studies over SDXL. "Rephrased prompt" stands for only rephrasing the text prompt without giving additional images. "Retrieve concepts" stands for using the missing concepts directly instead of using more detailed image captions for retrieval, and "Retrieve prompt" stands for using the prompt directly for retrieval. Best results are **bolded**.

| | ImageNet | | | CUB | | |
|---|---|---|---|---|---|---|
| | CLIP ↑ | SigLIP ↑ | DINO ↑ | CLIP ↑ | SigLIP ↑ | DINO ↑ |
| SDXL | $0.267 \pm 0.002$ | $0.136 \pm 0.001$ | $0.700 \pm 0.003$ | $\mathbf{0.315 \pm 0.001}$ | $0.172 \pm 0.003$ | $0.782 \pm 0.002$ |
| Rephrased prompt-SD | $0.266 \pm 0.002$ | $0.136 \pm 0.001$ | $0.705 \pm 0.003$ | $0.309 \pm 0.003$ | $0.170 \pm 0.002$ | $0.781 \pm 0.004$ |
| Retrieve concepts-SD | $\mathbf{0.274 \pm 0.001}$ | $\mathbf{0.141 \pm 0.001}$ | $0.702 \pm 0.003$ | $0.312 \pm 0.002$ | $0.173 \pm 0.002$ | $0.777 \pm 0.004$ |
| Retrieve prompt-SD | $\mathbf{0.274 \pm 0.001}$ | $0.140 \pm 0.001$ | $0.702 \pm 0.003$ | $0.314 \pm 0.001$ | $\mathbf{0.174 \pm 0.001}$ | $0.778 \pm 0.004$ |
| ImageRAG-SD | $\mathbf{0.274 \pm 0.001}$ | $\mathbf{0.141 \pm 0.001}$ | $\mathbf{0.709 \pm 0.002}$ | $0.314 \pm 0.001$ | $\mathbf{0.174 \pm 0.002}$ | $\mathbf{0.784 \pm 0.001}$ |

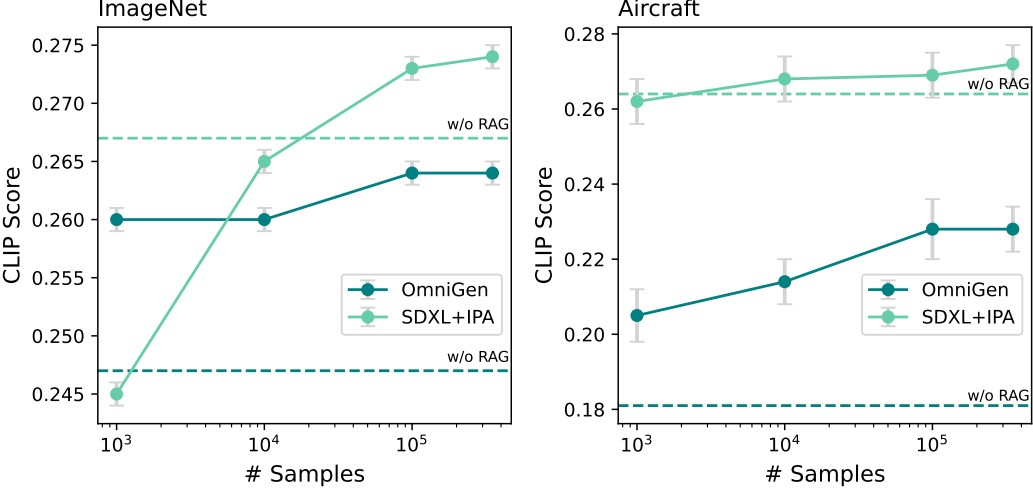

Figure 16: **Retrieval dataset size vs. CLIP score on ImageNet (left) and Aircraft (right).** Dashed lines represent the scores of the base models. Even relatively small, unspecialized retrieval sets can already improve results. More data leads to further increased scores. However, small sets may not contain relevant retrieval examples, and their use may harm results, particularly for stronger models.

### E.5 VLM ROBUSTNESS

We performed a VLM robustness experiment to choose which VLM we should use and make sure a VLM can accurately identify missing rare concepts in images. We randomly sampled 20 classes from the fine-grained dataset iNaturalist (Van Horn et al., 2018), and for each class, we generated 3 types of prompts: 1. "A photo of a <class_name>" 2. "A photo of a <class_name> and a <other>" 3. "A photo of a <other>". In total, we obtained 780 prompts; 20 prompts of the first type (1 per class), and 380 prompts for each of the second and third types (19 for each class, for every class other than the <class_name>). Finally, for each prompt, we asked the VLM if the prompt matches an image of that class, and if not, what are the missing concepts, as in our method. Note that each photo actually contains <class_name> but not <other>. This way, we were able to evaluate the ability of the VLM to identify missing rare concepts in images. The results of this experiment were a success; GPT-4o (Hurst et al., 2024), which is the VLM we used in our experiments, achieved 100% correct answers for the first and third prompt types and 99.7% correct answers for the second prompt type (1 wrong answer). We repeated this experiment with Gemini (Team et al., 2023) which achieved 95% correct first-type answers (1 wrong), 98.9% correct second-type, and 90% correct third-type, and with Qwen2.5-VL-7B-Instruct (Bai et al., 2025) which achieved 100%, 100%, and 92% correct answers, respectively. This indicates that while GPT identifies rare concepts best, both Gemini and Qwen also perform well and could potentially be used instead of GPT in the pipeline. We performed this experiment with LLaVA (Liu et al., 2023) as well, but it did not succeed at all (Simply answered 'No' on all queries).

### E.6 DIVERSITY

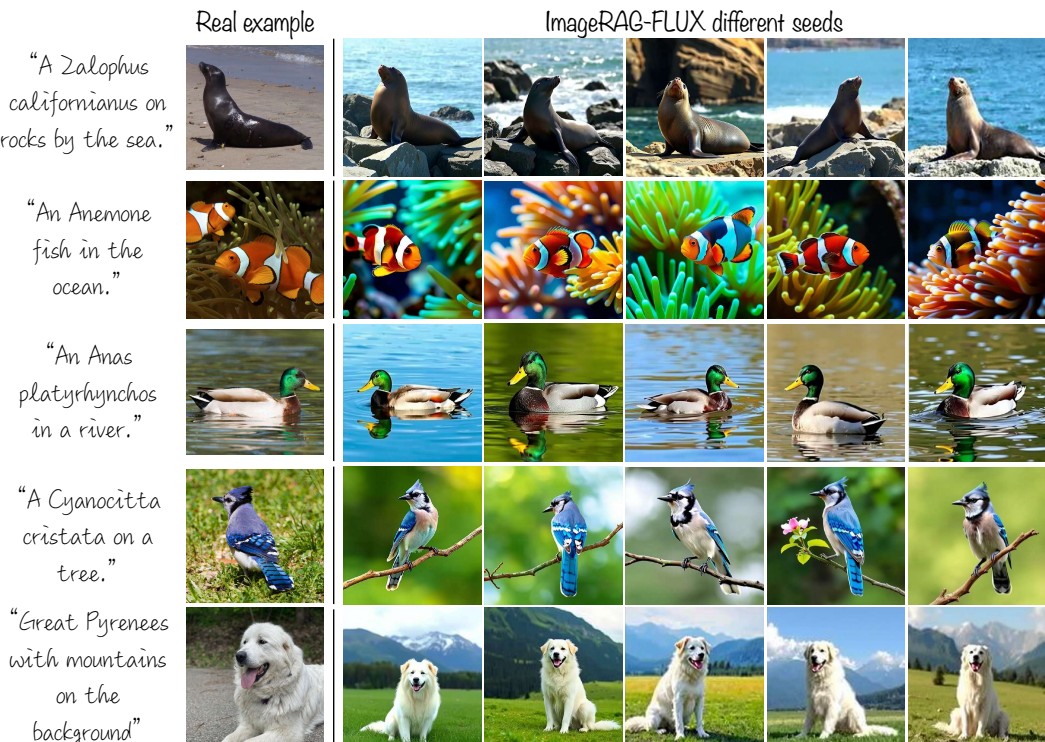

Figure 17: **ImageRAG-F (FLUX+OminiControl) generations with different seeds.** Left-most column presents a real example of the rare concept in the prompt, other columns present diverse generations of the same prompt by ImageRAG-F.

Generation examples with different seeds are presented in Fig. 18 (OmniGen), Fig. 19 (SDXL+IP-Adapter), and Fig. 17 (FLUX+OminiControl).

The diversity of generated results varies with the conditioning method, though all models can produce diverse outputs aligned with the text prompt. We observe some trade-off between diversity and

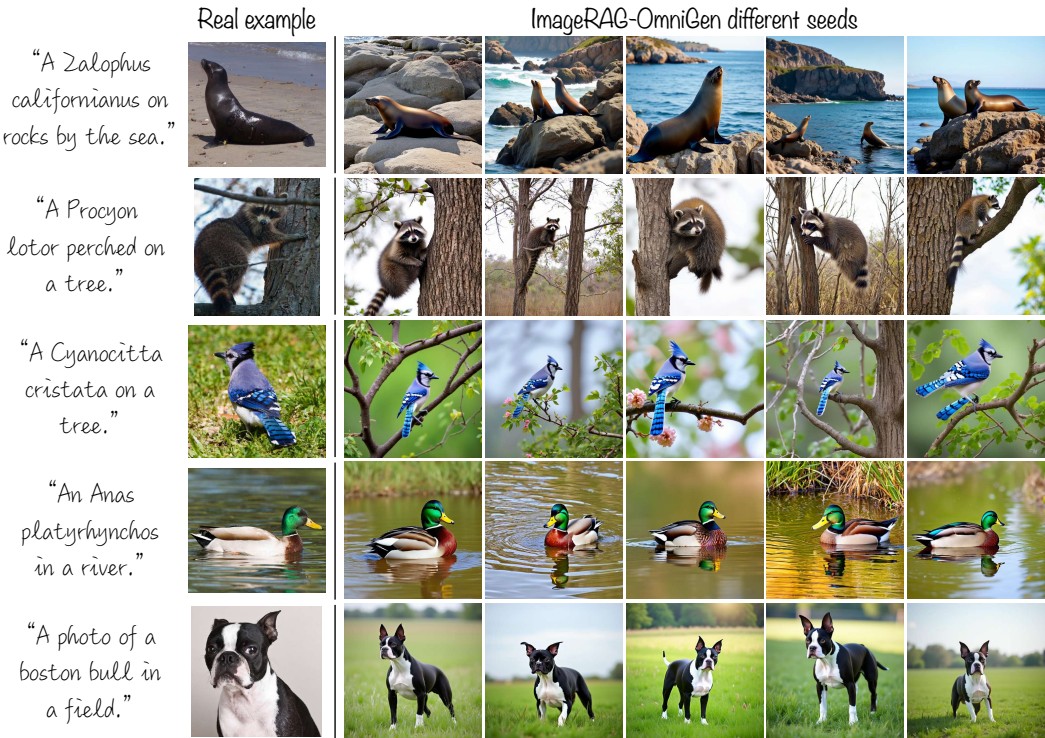

Figure 18: **ImageRAG-O (OmniGen)** generations with different seeds. Left-most column presents a real example of the rare concept in the prompt, other columns present diverse generations of the same prompt by ImageRAG-O.

textual faithfulness that depends on the chosen model. Since our approach is compatible with various architectures, the model can be selected based on the priorities of the user. For instance, SDXL+IP-Adapter yields outputs that are less diverse but closely match the reference image, OmniGen favors higher diversity at the cost of slightly reduced faithfulness, and FLUX+OminiControl provide a trade-off between the two.

## F   USER STUDY QUESTIONS

In the user study, we asked users to compare pairs of images at a time, by asking which one adheres better to the prompt and has better visual quality. We supplied real references (not from our dataset) for rare concepts with each pair. The questions we asked were: For each criteria, choose the better image out of A and B given the following text prompt: $<prompt>$. The less familiar concept "$<rare\_concept>$" is presented on the left of the image options.

- Better text alignment (choose A or B)
- Better visual quality (choose A or B)
- Overall preference (choose A or B)

Pair examples of using our method vs. other retrieval-based generation approaches can be found in Fig. 6. Due to lack of access to the models, all prompts and results of the other methods were taken from their papers.

## G   LLM USAGE

We have used an LLM (specifically, ChatGPT), for proofreading and rephrasing during paper writing.

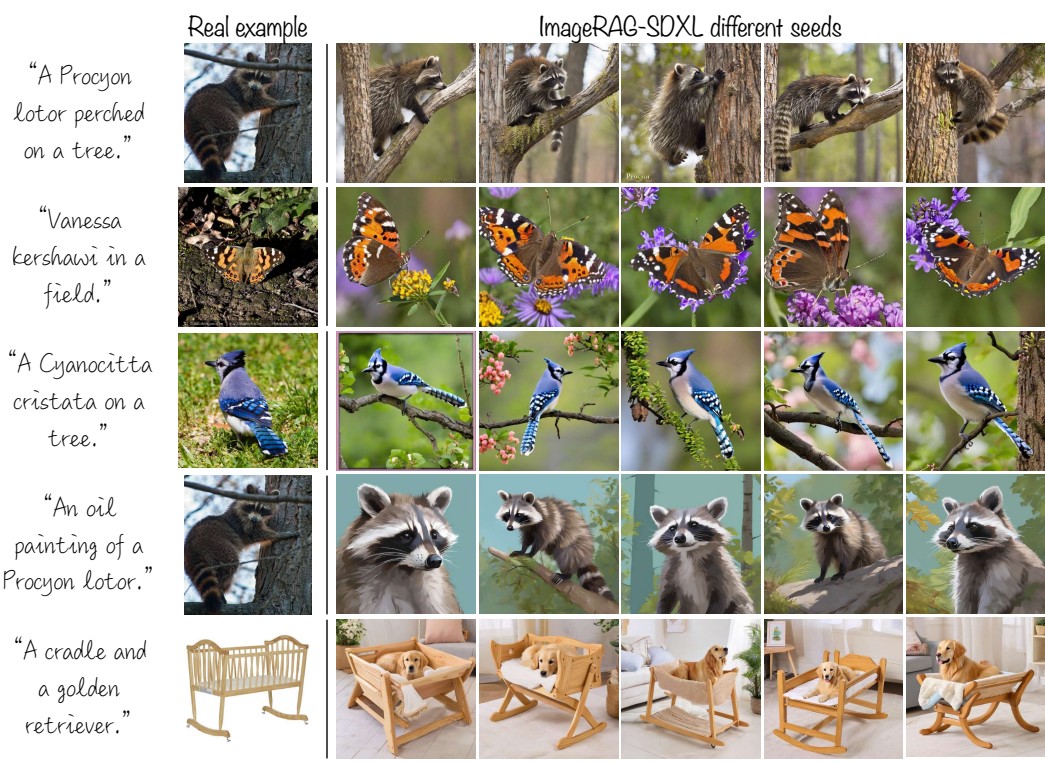

Figure 19: **ImageRAG-SD (SDXL+IP-Adapter)** generations with different seeds. Left-most column presents a real example of the rare concept in the prompt, other columns present diverse generations of the same prompt by ImageRAG-SD.

