# OpenReview forum: "ImageRAG: Dynamic Image Retrieval for Reference-Guided Image Generation"
_ICLR.cc/2026/Conference — ICLR 2026 Poster_

### Official Review · Reviewer_vzAE · 2025-10-27

**Soundness:** 2
**Presentation:** 3
**Contribution:** 2
**Rating:** 4
**Confidence:** 5

**Summary:**

This paper aims to address the recent generative models problems in generating rare or fine-grained concepts. ImageRAG framework is introduced as a training-free method for rare concept generation. ImageRAG identifies generation gaps with the guidance from a Vision Language Model. The VLM retrieves relevant images while ImageRAG uses them as context to guide the generation process.

**Strengths:**

1. The idea of applying RAG in image generation is up-to-date.

2. The method illustration is clear.

3. Writing is easy to follow.

**Weaknesses:**

1. As the most relevant work, RealRAG[A] is only mentioned in the related work section, while ignoring in the other sections, especially for the introduction and experiments.

a. Line-090: "previous works employing image retrieval for better image generation, train models specifically for the task, hindering wide applicability." The reviewer believe RealRAG does not train models specifically for task.

b. In the experiments, only several base models are included for comparison, more methods in this area should be compared, especially for RealRAG[A].

[A] Lyu Y, Zheng X, Jiang L, et al. RealRAG: Retrieval-augmented Realistic Image Generation via Self-reflective Contrastive Learning[C]//Forty-second International Conference on Machine Learning.

2. In the method, CoT is used for identifying challenging concepts. Could the author provide more discussion regarding this part? Is there a  trade-off between the CoT length and the generation performance? Is there any over thinking problem in this scenarios?

**Questions:**

### **1. On Related Work and Comparison with RealRAG**

**a. Conceptual Differences**

* In this paper, *RealRAG* (Lyu et al., ICML 2024) is mentioned only briefly in the related work section but not discussed elsewhere.
  Could the author elaborate on how the method differs conceptually from RealRAG, especially given that RealRAG also employs retrieval-augmented generation but does **not** require task-specific model retraining?
* What are the main methodological distinctions between ImageRAG and RealRAG in terms of:

  * Retrieval mechanisms and
  * Integration with the generation process?

**b. Experimental Comparison**

* In the experiments, only a few baseline models are included. Could the authors clarify why RealRAG and other recent retrieval-based image generation methods were excluded from the comparison?
* Have the authors evaluated the framework under the same experimental conditions as RealRAG (e.g., dataset, retrieval backbone, evaluation metrics) to ensure a fair comparison?

---

### **2. On Chain-of-Thought (CoT) Reasoning for Concept Identification**

* The authors mention that CoT reasoning is used to identify challenging or ambiguous concepts. Could the authors expand on how the CoT process interacts with the retrieval or generation stages?
* Is there a trade-off between **CoT length** and **generation quality** (e.g., longer CoTs improving reasoning but reducing visual fidelity or inference speed)?
* Have the authors observed any “overthinking” effects — cases where extended reasoning sequences introduce redundancy or semantic drift — and if so, how do the authors mitigate them?

---

> ### Author Response · Authors · 2025-11-21
>
> We thank the reviewer for their thoughtful feedback.
>
> Below we respond to your concerns:
>
> 1. **RealRAG:**
> * **Differences:** The key difference between RealRAG and ImageRAG is in our observation that discriminating between rare concepts is easier than generating them, and that the discrimination can be reliably performed by existing VLMs. Besides addressing the problem more directly, this approach also alleviates the need for training in general, let alone model specific training. RealRAG trains a retrieval method per model, while ImageRAG leverages the power of existing VLMs using a guided CoT process to retrieve images that the models do not know how to generate on their own, without imposing any training requirements.
> * **RealRAG training:** RealRAG requires training of a retriever module per generative model. This is what we meant in line 91, as clarified in the new paper revision.
> * **References:** We now reference RealRAG in the introduction as well and highlight our differences more clearly.
> * **Comparison:** the code and datasets used by RealRAG are not yet publicly available, hence we cannot compare with them directly.
> We ran ImageRAG over the datasets used in RealRAG (dogs, flowers, cars) and evaluated the results with CLIP-T, CLIP-I, which are the metrics used by RealRAG, and exhibited improvement over these datasets as well:
>
>
> |        | Dogs CLIP-T | Dogs CLIP-I | Flowers CLIP-T | Flowers CLIP-I | Cars CLIP-T | Cars CLIP-I |
> |--------------|-------------|-------------|----------------|----------------|-------------|-------------|
> | OmniGen      | 0.27        | 0.52        | 0.28           | 0.58           | 0.24        | 0.47        |
> | ImageRAG-O   | **0.30**        | **0.57**        | **0.31**           | **0.66**           | **0.29**        | **0.55**        |
> | FLUX         | 0.30        | 0.62        | 0.30           | 0.66           | 0.31        | 0.60        |
> | ImageRAG-F   | **0.31**        | **0.63**        | **0.31**           | **0.68**           | 0.31        | 0.60        |
> | SDXL         | 0.34        | 0.61        | 0.34           | 0.68           | 0.34        | 0.62        |
> | ImageRAG-SD  | 0.34        | **0.62**        | 0.34           | **0.69**           | 0.34        | 0.62        |
>
>
> We added these experiments with quantitative and qualitative results to **Appendix E.1** in the revised paper.
> For fair comparison, similarly to RealRAG, we included the dogs, flowers, and cars datasets in the retrieval dataset, as in our “proprietary data generation” experiment (**Appendix E.3**).
>
> Note that the range of CLIP scores RealRAG reported is significantly different from ours and from the scores reported in other papers (e.g. [1-5]). Specifically, RealRAG CLIP-T results range from 10-20, while the standard range is around 0.2-0.35. There is no code or explanation in the paper for how they calculate these metrics, so we wrote to the authors asking how they performed their evaluation, but so far we haven’t received an answer.
> Once we receive an answer, we would be happy to include both metric variants.
>
> [1] Scaling Rectified Flow Transformers for High-Resolution Image Synthesis (Esser et al., 2024)
>
> [2] OmniGen: Unified Image Generation (Xiao et al., 2024)
>
> [3] SDXL: Improving Latent Diffusion Models for High-Resolution Image Synthesis (Podell et al., 2023)
>
> [4] An Image is Worth One Word: Personalizing Text-to-Image Generation using Textual Inversion (Gal et al., 2022)
>
> [5] DreamBooth: Fine Tuning Text-to-Image Diffusion Models for Subject-Driven Generation (Ruiz et al., 2022)
>
> --------------
>
> 2. **CoT:** Our method utilizes a guided Chain-of-Thought (CoT), which we designed specifically to receive exact and relatively short answers, as specified in **section 3.1**. We start by asking the model to only answer Yes or No to the question if an initial image aligns with the text prompt. Next, if it answers no, we ask it to identify missing concepts in the image and use an example so it knows to only return short generic concepts as an answer. Finally, we ask it to generate image captions describing each of the missing concepts, and supply the captions only. These captions are used to retrieve the most relevant and helpful images to the model (explained further in **section 3.2**). The full prompts used in the CoT process are provided in **Appendix C**.
> By constraining the CoT answers to concise answers, we avoid overthinking and achieve accurate retrievals. Following your comments, we clarified and explained it better in the CoT part of our Method (**section 3.1**).
>
> We uploaded a revised version of our paper with changes and additions requested by the reviewers. For your convenience, changes are marked in blue.
>
> Thank you for your consideration.

---

> ### Comment · Reviewer_vzAE · 2025-11-25
> **Authors' responses address my concern.**
>
> After checking the authors' rebuttal, the reviewer's most concerns are addressed, the reviewer would like to increase score to support this paper to be accepted.

---

### Official Review · Reviewer_1GfW · 2025-10-28

**Soundness:** 3
**Presentation:** 4
**Contribution:** 4
**Rating:** 8
**Confidence:** 5

**Summary:**

The paper propose a image Retrieval-Augmented Generation (RAG) framework for image generation named ImageRAG. ImageRAG using VLM to retrieves relevant images to enhance custom image generation. Through extensive quantitative, qualitative, and subjective evaluation, paper show that incorporating retrieved references consistently improves the generation abilities of rare and fine-grained concepts.

**Strengths:**

1. The paper proposes a novel pipeline designed for image retrieval-augmented generation (RAG), which can retrieve relevant images to enhance the generative performance of diffusion models.
2. The proposed framework automates the common engineering practice of improving generation quality through reference images, demonstrating strong practical value.
3. The paper introduces an efficient approach for leveraging image knowledge database, which offers valuable insights for the development of multimodal large models.
4. The paper is clear and easy to follow.

**Weaknesses:**

1. If the target concept to be generated does not exist in the retrieval dataset, is there a possibility that irrelevant images may be selected, thereby potentially affecting the generation quality adversely? The paper should include corresponding quantitative analyses.
2. [Retrieval-Augmented Diffusion Models] has proposed a similar concept. The paper should provide a detailed comparison with this work with same base model and dataset, and analyze advantages of ImageRAG.
3. Will the diversity of generated images be constrained by the retrieval dataset, leading to visually similar outputs for same prompts? The paper should include relevant analyses to clarify this.

**Questions:**

SDXL+IPA have different image and text input interface, and cannot use the prompt template in Figure 3. How dose the ImageRAG handle models like this? Are the images correctly associated with the corresponding text snippets?

---

> ### Author Response · Authors · 2025-11-21
>
> We thank the reviewer for their constructive and helpful feedback, and for appreciating the novelty and practicality of our method.
>
> Below we respond to your concerns and questions:
>
> 1. **Irrelevant retrieval:** Please note that our method is applied only in cases where the VLM judges the initial image generated by the T2I model as not matching the prompt. As shown in our VLM robustness study (**Appendix E.5**), the VLM is very reliable in identifying such cases. Hence, if our model is applied, it is indeed only in cases where the initial model has failed.
> Having said that, as discussed in the limitations section, while it is possible that the retrieved image is entirely unrelated to the text (e.g., using a bird image for a dog prompt, **Figure 12** left), the model largely ignores the image in such cases and relies on the text, likely because unrelated concepts receive minimal cross-attention.
>
> 2. **Retrieval-Augmented Diffusion Models (RADM):** While this work has a similar idea to ours, there are some key differences between our works. First, they train for the RAG task specifically, while we do not require any training. Second, they retrieve examples similar to the text prompt, while we retrieve examples that are similar to missing concepts using our guided CoT process. As shown in our ablation studies (**Section 4.3**), this strategy yields better results compared to retrieving images similar to the prompt directly. As RADM train for the task, we cannot compare to them using their model as a base model directly, however in our user study, we compared our results to their results using images from their paper. We have now highlighted the differences between our works in the revised paper version.
>
> 3. **Diversity:** We discuss diversity in **Appendix E.6** and supply visual examples of all models with different seeds. As shown and discussed, the diversity of the generated results varies with the conditioning method, however all models can produce diverse outputs aligned with the text prompt.
>
> 4. **SDXL+IPA conditioning prompt:** While the conditioning is different, we discovered that we can use the same prompt template, and SDXL still adheres to it. As mentioned in our implementation details section (**Appendix A**), for SDXL we only used 1 image, as we used an implementation of IPA that only allows 1 image.
>
> We uploaded a revised version of our paper with changes and additions requested by the reviewers. For your convenience, changes are marked in blue.
>
> Thank you for your consideration.

---

### Official Review · Reviewer_fN6i · 2025-10-31

**Soundness:** 2
**Presentation:** 3
**Contribution:** 2
**Rating:** 4
**Confidence:** 3

**Summary:**

This paper introduces RAG into text-to-image generation process for some rare or fine-grained concepts. This approach is training-free and model agnostic, and can operate with off-the-shelf image-conditioning interfaces. Image RAG contains four steps: 1/ rough image generation from a text prompt; 2/ use VLM to identify the missing concepts in a CoT process,  3/ retrieve reference images corresponding to the missing concepts from an external dataset; 4/ regenerate the image using these reference images by a generative model. The authors evaluated the method using three different base models, including SDXL, FLUX, and OmniGen, on three benchmarks.

**Strengths:**

- The adaptation of RAG to text-to-image generation is inspiring. It focuses on "polish" the generated image at inference time rather than retraining.

- The formulation of CoT-based retrieval is promising, which is effective for identifying the missing visual elements.

- The evaluation is comprehensive, including qualitative comparisons, user study, and ablation studies. It is good to see that failure cases are included in this paper, which help the reader to better understand the limitation of this paper.

**Weaknesses:**

- The applicable scenario is limited. ImageRAG suits most for the scenario where rare or even weird concepts exist, such as "a boston bull", but the successfulness greatly depends on the coverage of the retrieval dataset (e.g. LAION-350K subset). It would be better to see that this method can handle well with the complex and lengthy prompts that are usually more customer-driven.

- No report or measurement of the latency and computational cost of this multi-step pipeline, which is even more important than the metrics measuring image quality. We should always consider the latency for real-time or interactive use cases.

- The technical novelty is limited. Even though the concept of ImageRAG is good, each component in this framework is individually normal. And the success of ImageRAG heavily rely on the performance of the submodule such as CLIP, GPT-4o and T2I, etc.

- The performance improvements in CLIP or DINO metrics are in general very limited (<0.01 absolute).

**Questions:**

- Since the limitations are listed in this paper, is there any solution or strategy to improve them?

- Is there any failure cases that VLM misidentify the gaps between text prompts and rough images?

---

> ### Author Response · Authors · 2025-11-21
>
> We thank the reviewer for their thoughtful review and for recognizing the strength of our RAG adaptation and the comprehensiveness of our evaluation.
>
> Below we respond to your concerns and questions:
>
> 1. **Applicable scenarios:** While our primary focus is rare-concept generation, our method is applicable beyond this setting. It supports creative generations (**Figure 10**) and personalized or brand-specific content generation where the user’s private images serve as a high-quality retrieval set (**Figure 14**).
> Additionally, following your suggestion, we performed an experiment testing longer and more complex prompts such as “A raccoon wearing formal clothes, wearing a tophat and holding a cane. The raccoon is holding a garbage bag. Oil painting in the style of pointilism.” and included visual results in the revised paper version in **Figure 11**. While long prompts might confuse the models and still omit some concepts from the generated image, ImageRAG improves the overall generation and helps models generate more of the prompt.
>
> 2. **Latency report:** Latency measurements were reported in the implementation details (**Appendix A**) for GPT-4o as the VLM. We observed similar end-to-end latency with Qwen-2.5 as the VLM and added it to the implementation details of the revised version. We note that adding additional reference images does not lead to a meaningful increase in the generation time of the T2I model.
> Since retrieval embeddings are pre-computed, retrieval itself is sub-second, and the only overhead is the short CoT reasoning step (10-30 seconds in total) and the time required for generating the additional image, which depends on the base generative model. We clarified it in the revised paper.
>
> 3. **Novelty:** Although ImageRAG is built from standard components, the contribution lies in how these components are orchestrated into a plug-and-play pipeline requiring no training. We observe that VLMs are powerful and more reliably and accurately identify missing and rare concepts in images, even when image generation models cannot generate them. Inspired by that observation, we leverage existing VLMs and design a guided CoT process to retrieve the most relevant and helpful images that guide image generation, enabling the models to generate concepts they couldn’t generate before.
> This design yields consistent improvements across architectures and datasets, which we argue is a substantive practical and methodological contribution.
>
> 4. **Performance improvements:** When looking at the CLIP/DINO scores alone, improvement seems relatively low. However, it’s because these metrics measure coarse semantic similarity: two bird species are close in embedding space even though they differ meaningfully for concept-specific generation. Improvements are much clearer when looking at the GPTScores, user studies, and visual results, which more faithfully capture the intended evaluation signal. We added this clarification to the revised paper.
>
> 5. **Solutions to limitations:** As noted in **section 3.2**, our method allows using a retrieval threshold that ensures only very similar examples influence generation. This mitigates noise and dataset mismatches. Incorporating user-specific datasets further improves robustness and relevance. Moreover, due to the flexibility of our method, whenever new similarity metrics, feature extractors, T2I, or conditioning models arise, we can replace the retrieval method or base model used with newer ones to improve retrieval and generation quality, without any training or modifications.
>
> 6. **VLM failure cases:** The VLM may sometimes misidentify an initial image as a match to the original prompt, especially if the prompt could be interpreted in multiple ways. For example, for the prompt “A photo of a love in the mist”, OmniGen generated an initial image of a couple in the mist. This does match the prompt, hence the VLM identified it as correct; however, our intention was the flower “love in the mist”. In that case, we can either add “the flower” to the prompt, or apply our method by including the fact that the image and prompt do not align directly in the query prompt, so the VLM knows it should refer to the other meaning. We added this example to the revised paper. However, as demonstrated in the VLM robustness experiment we performed (**Appendix E.5**), the VLMs are very reliable and accurate in identifying rare concepts in images and succeed in most cases.
>
> We uploaded a revised version of our paper with changes and additions requested by the reviewers. For your convenience, changes are marked in blue.
>
> Thank you for your consideration.

---

> > ### Comment · Reviewer_fN6i · 2025-11-27
> >
> > Thanks for addressing most of concerns carefully. I believe the revision significantly strengthens the paper. The method is now better motivated, better evaluated, and more clearly positioned. I am inclined to raise my score and recommend acceptance.

---

### Official Review · Reviewer_bsqR · 2025-10-31

**Soundness:** 3
**Presentation:** 3
**Contribution:** 3
**Rating:** 4
**Confidence:** 3

**Summary:**

This paper proposes ImageRAG, a training-free retrieval-augmented generation method that enhances text-to-image models by dynamically identifying generation gaps using a vision-language model and retrieving relevant reference images to guide the synthesis of rare or fine-grained concepts. The approach is model-agnostic, demonstrated through evaluations on diverse backbones like OmniGen, SDXL, and FLUX, showing improved alignment with prompts without requiring additional training.

**Strengths:**

The method's adaptability is a key strength, as it seamlessly integrates with existing image-conditioning models like IP-Adapter and OminiControl, enabling broad applicability across different architectures. Extensive experiments, including quantitative metrics, qualitative examples, and human studies, robustly validate that ImageRAG consistently improves rare concept generation, with users preferring it over baselines in text alignment and visual quality.

**Weaknesses:**

1. The method's efficacy is highly dependent on the retrieval dataset; if it lacks relevant images (e.g., specializing in birds when generating dogs), performance may not improve, as illustrated in the retrieval data limitations.

2. It relies on the VLM's accuracy for gap identification; errors in concept detection (e.g., false positives in alignment checks) could lead to missed enhancements, though the paper notes robustness issues with some VLMs.

3. While the paper notes VLM API calls add 10–30 seconds, it lacks comparisons to baseline T2I inference latency and does not explore optimizations (e.g., cached embeddings for small datasets) for real-time use cases.

4. Although diversity is mentioned, the paper does not quantify how ImageRAG impacts generation diversity across models (e.g., SDXL’s low diversity vs. OmniGen’s high diversity) or whether retrieval introduces bias toward reference image styles.

**Questions:**

See weakness

---

> ### Author Response · Authors · 2025-11-21
>
> We thank the reviewer for their detailed and valuable feedback and for recognizing the adaptability and robustness of ImageRAG.
>
> We respond to your concerns below:
>
> 1. **Retrieval dataset:** Indeed, as mentioned in our limitations section, our method depends on the retrieval dataset. However, in many realistic scenarios, relevant retrieval datasets do exist. For example, as shown in the paper, a generic dataset such as LAION could help with general or rare concept generation.
> Moreover, as demonstrated in **Appendix E.3** (proprietary data generation), our method could help even further when user-specific datasets are available, e.g., when using a private collection of images, and for personalized generation, e.g., for generating a personal concept or a company brand, as demonstrated in **Figure 14**.
>
> 2. **VLM robustness:** One of the core observations of our paper is that VLMs’ ability to identify rare concepts is better than the generation abilities of such concepts by generative models. As exemplified in our VLM robustness experiment (**Appendix E.5**), VLMs and in particular GPT-4o, Qwen, and Gemini are capable of identifying rare concepts in images with a very high accuracy and therefore are more reliable.
>
> 3. **Latency:** Our experiments show that adding reference images to the base generation models does not meaningfully increase the latency of the models.  Therefore, the only additional time our method requires is the time required for the CoT process, which is about 10-30 seconds in total, whether using GPT or Qwen as the VLM, and the time required for generating the second image, which depends on the base generative model and could be optimized per model. We clarified this point in the implementation details section (**Appendix A**) of the revised paper. Regarding cached embeddings, please note that we compute the dataset embeddings only once and therefore finding compatible reference images at inference operates in sub-second time, as mentioned in the implementation details.
>
> 4. **Diversity:** As discussed in **Appendix E.6** (Diversity), and shown in **Figures 17-19**, ImageRAG is capable of producing diverse results that are faithful to the text prompt with all base models. It is hard to quantify the diversity compared to the base models, because while they may be more diverse superficially, they fail to generate the target concepts. ImageRAG ensures accuracy and relevance, which is our priority.
> Regarding bias toward reference images, the general style of the output image often differs significantly from the style of the reference image, and between generated images with the same prompt and reference image, and different seeds. For example, in **Figure 17**, each image of the prompt “A Cyanocitta cristata on a tree” has a different orientation of the bird, a different background, and different colors. **Figure 19** presents two prompts with the concept “Procyon lotor” with very different styles, which in turn also yield very different images and styles, even when the reference is the same.
>
> We uploaded a revised version of our paper with changes and additions requested by the reviewers. For your convenience, changes are marked in blue.
> Thank you for your consideration.

---

### Author Response · Authors · 2025-11-21
**General comment to all reviewers**

Dear reviewers,

Thank you all for your constructive feedback and helpful suggestions.

We have uploaded a revised version of the paper incorporating your comments, with all additions marked in blue for clarity.

Below, we summarize the main changes and additions:
* **New experimental results** over the Flowers, Dogs, and Cars datasets (**Appendix E.1**).
* **Additional visual results** for longer and more complex prompts (**Figure 11**).
* **Clarifications** were added, including for the differences between our method and previous work, latency and running times, method and evaluation metrics explanations.

We additionally respond to each reviewer individually.

Thank you for your time and consideration.

---

### Author Response · Authors · 2025-11-30
**AC summary**

Dear AC,

We appreciate your time and effort, and summarize the rebuttal discussion for your convenience. We start with the current status and follow with the main concerns that were raised during the rebuttal, along with our responses to them.

**Status:**

Reviewer 1GfW had initially rated our paper with 8 (accept).

As we have addressed the concerns of reviewers vzAE and fN6i, they both decided to raise our scores towards acceptance (raised from 4 to 6), as was mentioned in their comments prior to the leakage notification. We were not aware of the leakage until it was announced, and are afraid the decision to revert all scores will unfairly influence our submission. We hope that the AC can go over the rebuttals and validate that we indeed resolved the concerns of vzAE and fN6i, and thus accept their revised assessment.

Finally, reviewer bsqR hasn’t answered yet, but we hope we managed to address all of their concerns as well.

During the rebuttal, we uploaded a revised version of our paper with changes and additions requested by the reviewers. Changes are marked in blue for your convenience.
The main changes and additions to the paper were:
* **New experimental results** over the Flowers, Dogs, and Cars datasets (**Appendix E.1**).
* **Additional visual results** for longer and more complex prompts (**Figure 11**).
* **Clarifications** were added, including for the differences between our method and previous work, latency and running times, method and evaluation metrics explanations.

**Main discussion points:**

* **Diversity** - reviewers bsqR, 1GfW asked about the diversity of our method. We pointed them to the discussion and visual results about diversity in **Appendix E.6** in our paper and explained that, as shown and discussed, the diversity of the generated results varies with the conditioning method; in any case, all models produce diverse outputs aligned with the text prompt.
* **Latency** - reviewers bsqR, fN6i asked about the latency of our method. We extended **Appendix A** of our paper, which contained latency measurements.
* **VLM robustness** - reviewers bsqR, fN6i asked about the robustness of the VLMs used in our method. We pointed them to the VLM robustness experiment we performed (**Appendix E.5**), showing that VLMs reliably identify rare concepts in images with high accuracy.
* **Comparisons with prior work** - reviewer 1GfW asked about comparison and differences between our method and Retrieval-Augmented Diffusion Models (RADM), and reviewer vzAE similarly asked about RealRAG. To both, we answered that first, these methods require training, while our model does not. Moreover, we further explained our differences and highlighted them better in the paper. For RADM, we mentioned that in our user study (**Section 4.1.1**), we compared our results to their results using images from their paper. For RealRAG, we ran our method over the datasets they used and added the results to the paper (**Appendix E.1**), showing improvement in them as well across all models.
* **Applicability** - reviewers bsqR, fN6i asked about the applicability of our method given that it requires a relevant retrieval dataset. We explained that in many realistic scenarios, relevant retrieval datasets do exist, as shown in the paper. Moreover, we pointed them to **Appendix E.3** that demonstrates our method could help even further when user-specific datasets are available, e.g., when using a private collection of images, and for personalized generation, e.g., for a personal concept or a company brand, as demonstrated in **Figure 14**. Additionally, to answer reviewer fN6i who asked about more complex and long prompt generation, we performed an experiment testing longer and more complex prompts and included our results in **Figure 11** in the revised paper.

We highly appreciate your consideration.
Thank you.

---

### Meta-Review · Area_Chair_wu5E · 2026-01-07

**Summary:**

The original review scores of this paper are 4 (reviewer vzAE), 4 (reviewer fN6i), 4 (reviewer bsqR), 8 (reviewer 1GfW).

Reviewer vzAE has major concerns about comparison to RealRAG, and more discussion on CoT. Reviewer fN6i has major concerns about applicable scenarios, latency, novelty, some experiment results, and limitation/failure-cases. reviewer bsqR has major concerns about dependency on the retrieval dataset and VLM, inference latency, and diversity.

**Reviewer Concerns:**

Most concerns from reviewer vzAE and fN6i have been resolved, and both agreed to raise the score to accept. Concerns of bsqR are mostly resolved too.

**Reviewer Scores:**

reviewer vzAE and fN6i would raise the score to accept. Reviewer should also raise bsqR too.

---

### Decision · Program_Chairs · 2026-01-26

Accept (Poster)